# Structure of giant kelp Photosystem I-FCP uncovers drivers of antenna evolution across the red lineage

Jenevieve D. Weissman[1,5], Pablo Maturana [1,2,5], Hui M. O. Oung[1], Reece Riddle [1,3], Gabrielle Wyatt [1], Viktoria G. T. Dubinin[1], Philipp Zerbe [1] & María Maldonado [1,4] ✉

Brown algae and other red-algae-derived organisms are major contributors to global CO$_2$ fixation via photosynthesis. To understand the photosynthetic function of brown algae, we obtained the structure of giant kelp *Macrocystis pyrifera* photosystem I (PSI) with a fucoxanthin-chlorophyll-protein (FCP) antenna and compared it to known structures from the red-algal lineage. We identified differences in *M. pyrifera*'s antenna composition, architecture, and chlorophyll networks, as well as a pronounced variation in transmembrane hydrophobic thickness across the PSI-FCP supercomplex, with implications for photochemical function. Our work lays the foundation to understand kelp's high photosynthetic productivity and reveals drivers of antenna conservation and diversification for the red lineage.

The photosynthetic conversion of inorganic carbon into organic material, i.e., primary production, sustains most life on this planet[1]. Roughly half of the global primary production originates from photosynthesis in the oceans, carried out by a range of prokaryotic and eukaryotic organisms[2]. A diverse group of eukaryotic phototrophs derived from red algae ("red lineage") is a major contributor to primary production, marine ecosystem maintenance and coastal economic activity around the world[2–5]. Understanding the molecular mechanisms of photosynthesis and how it has diversified across the red lineage will illuminate this critical metabolic process and enable the development of new ocean-based environmental strategies. Here, we examine the structural features of the photosynthetic complex photosystem I (PSI) of brown algae to shed light on its specialization and diversification, as well as on the evolutionary relationships between the different clades of the red lineage.

Oxygenic photosynthesis uses solar energy to convert carbon dioxide (CO$_2$) and water into organic molecules and molecular oxygen. The photochemical reactions of photosynthesis are enabled by multiprotein photosystems I and II (PSI, PSII) in the thylakoid membrane, which harness light to power the transfer of electrons from water to NADP$^+$ and pump protons into the thylakoid lumen. PSI catalyzes the transfer of electrons from electron donors such as plastocyanin (Pc) and cytochrome $c_6$ to the electron acceptor ferredoxin (Fd). The synthesis of NADPH and ATP resulting from the light reactions of photosynthesis drives the fixation of carbon from CO$_2$ into organic molecules[1]. Photosystems absorb light by the excitation of chromophores such as chlorophylls (chl). To increase the efficiency of light absorption and to obtain protection against photo-oxidative damage, in many organisms photosystems associate with additional light-harvesting proteins that form a multi-protein "antenna"[1]. The antenna funnels the absorbed light into the photosystem reaction center by transferring the energy across the chromophores in a process known as excitation energy transfer (EET)[1]. Whereas photosystems are structurally and functionally conserved, their antenna systems and chromophores are highly diverse, tuned to absorb light in different environments[6]. The most abundant antenna system in eukaryotic phototrophs, including the red lineage, is that composed of light-harvesting complex (LHC) membrane proteins. The LHC family has

[1]Department of Plant Biology, University of California-Davis, Davis, CA, USA. [2]Present address: Departamento de Bioquímica y Biología Molecular, Universidad de Concepción, Concepción, Chile. [3]Present address: National Cancer Institute, Frederick, MD, USA. [4]Present address: Monash Biomedicine Discovery Institute, Monash University, Clayton, VIC, Australia. [5]These authors contributed equally: Jenevieve D. Weissman, Pablo Maturana. ✉e-mail: maria.maldonado@monash.edu

diversified into subfamilies that share a three-transmembrane-helix topology but can differ in the number and types of their bound chromophores[6,7]. The LHC diversification is salient in the red lineage, where the Lhcr subfamily present in red algae expanded and evolved into at least six different subfamilies in subsequent clades[6,7]. Additionally, LHCs in the red lineage have been historically referred to by their bound chromophores, e.g., fucoxanthin-chlorophyll *a/c* protein (FCP), regardless of their LHC subfamily[6,7].

Red algae, as well as plants and green algae, originated from the engulfment of photosynthetic cyanobacteria. Through gene transfers and other processes, this primary endosymbiont evolved into a stable organelle, i.e., the chloroplast, of the red alga[1]. The red lineage includes the diverse phyla Cryptophyta, Haptophyta, Ochrophyta (photosynthetic stramenopiles such as brown algae and diatoms) and Myzozoa (e.g., dinoflagellates). In these phyla, higher-order endosymbiosis of a red alga or another red-lineage organism led to the formation of increasingly complex chloroplasts upon engulfment and stabilization through gene transfer. The evolutionary trajectory of the set remains elusive, with phylogenomic support for various models of serial or parallel endosymbioses[8–10] (Supplementary Fig. 1a-d). Elucidating the evolutionary relationships between the red-lineage phyla and the evolution of their photosystems and antennae is key to understand this critical group of primary producers.

The comparative study of protein structure and subunit composition provides a complementary approach to disambiguate evolutionary relationships between organisms, especially with multi-protein complexes involved in core metabolism, such as PSI[11,12]. Currently, high-resolution structures of PSI with its antenna assembly are available for various red algae, cryptophytes, haptophytes and dinoflagellates[13–20]. Among ochrophytes, structures are only available for diatoms, organisms of the Diatomista clade[21–24] (Supplementary Fig. 1e). Although diatoms are major marine phototrophs and well-studied model systems, their features should not be taken as representative of all ochrophytes[23]. The structural study of a Chrysista ochrophyte, e.g., a brown alga such as *Macrocystis pyrifera* (giant kelp), would resolve questions about the generalizability of features across the ochrophyte phylum and the red lineage.

Kelp forests are critical components of coastal ecosystems and economies. With a net primary productivity up to three times higher than marine phytoplankton, kelp forests are the most net-productive marine biome per unit area[3–5]. Thus, kelp could provide significant additions to ocean-based carbon sequestration[25,26]. Understanding the structural details of brown algal photosynthesis is fundamental to elucidate kelp's superior net productivity, to inform red-derived plastid evolutionary models and to develop applications that leverage kelp's photosynthetic abilities. To begin to tackle these questions, we determined the structure and composition of a PSI-FCP antenna supercomplex of the brown alga *Macrocystis pyrifera* (giant kelp).

Our structural analyses in this work identified patterns across PSI-FCPs of red-lineage organisms, including membrane-thickness variation and differences between Chrysista and Diatomista EET pathways, with implications for photosynthetic function. We propose that contingent protein:protein interactions, gene loss, hydrophobic-mismatch management and neutral evolution are key drivers of antenna FCP subfamily conservation and diversification across the red lineage.

## Results
### *M. pyrifera* PSI-FCP structure and FCP phylogeny

After optimizing protocols to decrease the viscosity of the kelp sample, we isolated a chloroplast-enriched fraction from fresh *M. pyrifera* blades. Using digitonin and sucrose gradients, we purified PSI-FCP for structural determination with cryogenic electron microscopy (cryoEM) (Supplementary Figs. 2–6, Supplementary Tables 1–3). We observed a PSI-FCP supercomplex with 11 PSI subunits (PsaA/B/C/D/E/F/I/J/L/M/R) and 18 FCP subunits arranged in two belts (first belt: FCP1-11; second belt: FCP13/15/16/17/19/A/B) (Fig. 1a, b). PSI subunits were highly conserved with respect to the red lineage. The structure also confirmed the lack of PsaK in brown algal PSI (present in red algae, cryptophytes and haptophytes), as well as the lack of PsaO (present in red algae and cryptophytes)[13–24,27,28]. The Supplementary Note 1 and Supplementary Fig. 7 provide more details on the *M. pyrifera* PSI-FCP structure, electron transfer and protein-protein interactions.

FCPA/B are antenna positions not previously seen in other organisms[13–24,27,28]. Additionally, a purification with different detergent conditions showed *M. pyrifera* PSI-FCP supercomplexes with larger antennae suggestive of additional positions in the vicinity of FCPA/B (Supplementary Fig. 8a-f). The size and shape of these larger *M. pyrifera* antennas were different from those of the currently largest antennae of diatoms or haptophytes[18,21] (Supplementary Fig. 8g-i). However, the resolution of that dataset was insufficient for atomic modelling of these larger antennae.

We identified *M. pyrifera*'s FCP complement and determined its phylogenetic relationships (Supplementary Fig. 9, Supplementary Data 1). We resolved *M. pyrifera*'s FCPs into seven subfamilies (Lhcr, Lhcf, Lhcq, Lhcx, Lhcz, CgLhcr9-like and RedCAP), further establishing that this classification is robust across the red lineage[29,30]. We observed substantial species-specific expansion of subfamilies Lhcf and Lhcx (Supplementary Fig. 9). Lhcx proteins play photoprotective roles in the energy-dependent component of non-photochemical quenching via PSII in other ochrophytes[31]. Given that *M. pyrifera* does not use this type of photoprotection[32,33], the presence of eight *Lhcx* genes suggests they may have alternative functions in brown algae.

All FCP subfamilies except for Lhcx and Lhcz were present in our structure (Fig. 1d). On the first belt, we assigned Lhcr, Lhcq, CgLhcr9-like (Cg9-l) and RedCAP subunits with high confidence. On the second belt, we assigned two subunits as Lhcr (FCP13) and Lhcf (FCP17) with high confidence. The resolution of six FCP focused-refined maps was insufficient to assign protein sequence (FCP2/15/16/19/A/B). Nevertheless, FCP subfamilies are known to show structural differences[6] (Supplementary Fig. 10a–e). For instance, Lhcf proteins have a shorter and straighter helix C than Lhcr and Lhcq proteins, providing useful features to differentiate subfamilies (Supplementary Fig. 10f–j). Thus, we reasoned it would be possible to identify the best subfamily match by calculating map-model-fit (Q-scores, Supplementary Fig. 10k–s)[34]. Q-scores of assigned models and maps of different subfamilies fit with non-cognate partners showed that subfamily discrimination is possible even at -5 Å resolution (Supplementary Fig. 10k, l). Thus, we measured the Q-scores of assigned subfamily model representatives into each unassigned map (Supplementary Fig. 10m–r) and of our unassigned poly-ala models into representative maps of each subfamily (Supplementary Fig. 10s). These complementary measures identified Lhcf as the best-matching subfamily for FCP15/16/19/A, given our current data, to be confirmed with higher resolution structures. The subfamilies for FCP2/B remained unassigned due to their poorly discriminating Q-scores.

To understand drivers of antenna conservation and diversification and shed light on evolutionary relationships, we compared antenna architecture and FCP subfamily composition across the red lineage (Supplementary Fig. 11). We followed the nomenclature of *Chaetoceros gracilis* (diatom), the largest reported ochrophyte antenna, to minimise re-naming (see Supplementary Fig. 12 for conversion)[21]. Note that the FCP number refers to the position in the antenna, not to the gene name. For clarity, all antenna proteins are referred to as "FCP" using our numbering, even if the LHC subunit binds other chromophores in certain organisms.

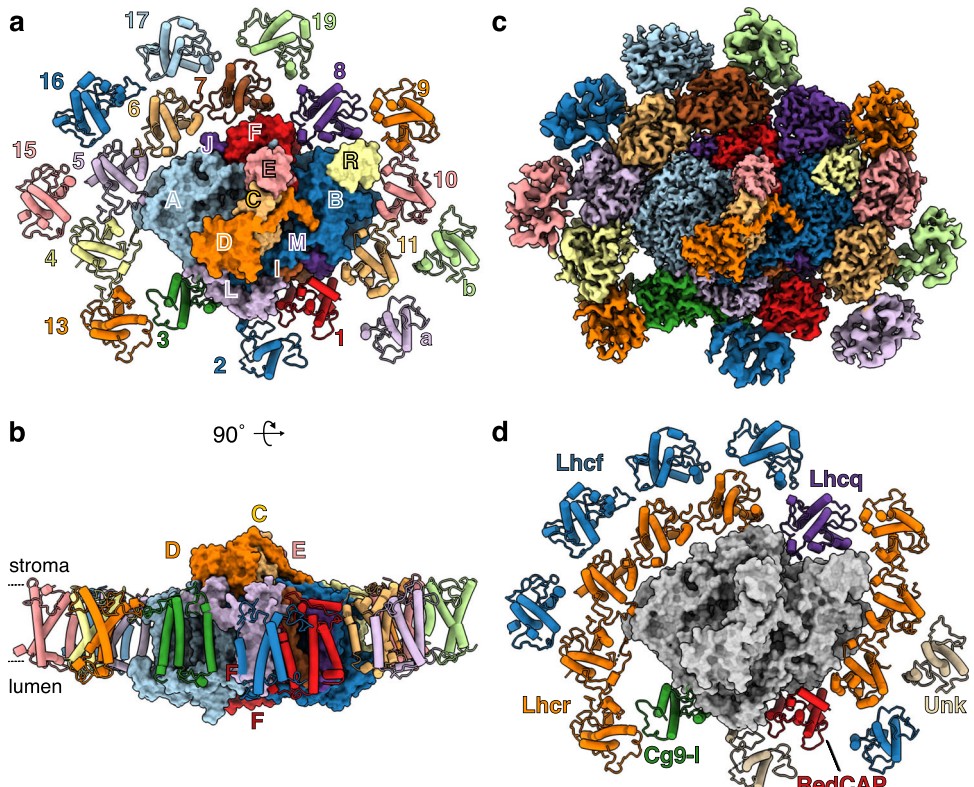

**Fig. 1 | Structure of *Macrocystis pyrifera*'s PSI-FCP supercomplex.** Atomic model of *M. pyrifera* PSI-FCP supercomplex viewed from the stroma (**a**) or the membrane (**b**). PSI in surface, FCP proteins in cartoon, coloured by subunit. The approximate locations of the stroma and lumen are shown with dashed lines. For clarity, "Psa" and "FCP" suffixes are omitted from subunit labels (e.g., PsaA labeled A, FCP labeled 1). To prevent confusion with PsaA/B, FCPA/B are labeled in small print. **c** Stromal view of cryoEM composite map of *M. pyrifera* PSI-FCP supercomplex, coloured by subunit as in (A-B). **d** Stromal view of *M. pyrifera* PSI-FCP supercomplex labelled by FCP subfamily. PSI shown in grey; Lhcr in orange; RedCAP in red; CgLhcr9-like (Cg9-l) in green; Lhcq in purple; Lhcf in blue; unknown family (Unk) in beige. FCP numbers used in this work shown, with FCP suffixes omitted for clarity.

## Conserved protein interactions and chromophore binding sites differentiate the 1st and 2nd belts

Similarities in PSI-FCP sequence, subunit composition and architecture are strongest in PSI, weakening with increasing distance from the core across the red lineage (Supplementary Fig. 11, Supplementary Text). The Lhcr majority that is seen in *M. pyrifera*'s first belt is conserved across all the red lineage; this Lhcr majority is absolute in red algae and cryptophytes[13–24] (Supplementary Fig. 11, Supplementary table 5). We conclude that the conservation of the first belt is driven by the sequence conservation of the PSI subunits, to maintain the PSI:FCP binding interactions. The Lhcr first-belt majority is further aided by subfamily-specific features, e.g., Lhcrs' aromatic residues that allow for FCP:FCP interactions not seen in other subfamilies (Supplementary Fig. 13a–e). Lhcr proteins in the first belt establish hydrophobic interactions between aromatic residues on the N-terminus and the C-A loop of the binding partners (Supplementary Fig. 13a-c). These aromatic-mediated interactions are specific to the Lhcr family and conserved across organisms[13–24]. Lhcr proteins are also well suited to the first belt thanks to their additional family-specific chromophore binding site (chl 415)[29] (Supplementary Fig. 13d, e). This Lhcr-specific chlorophyll is ~6 Å away from chl 407 of the adjacent FCP, poising these chlorophyll molecules for efficient excitation energy transfer (EET) into PSI's reaction center[35,36]. Indeed, fast transfer (<10 ps) between these chlorophylls has been measured in dinoflagellates[20]. Thus, replacing Lhcr proteins in the first belt with other subfamilies leads to changes to the EET network.

In contrast to the first-belt Lhcr majority, our structural analyses suggest that the majority of *M. pyrifera*'s second belt is composed of Lhcf proteins (Supplementary Fig. 10k-s). The exception is FCP13,

which is Lhcr. This Lhcf-majority composition of the second belt would be unlike most other red-lineage PSI. In diatoms and haptophytes, second-belt FCPs are predominantly Lhcq[21–24,29], whereas in cryptophytes the second belt is unanimously Lhcr[15,16] (Supplementary Fig. 11). However, dinoflagellates have a similar pattern to *M. pyrifera* in which all second-belt FCPs are classified as subfamily Lhcf[19,20] (Supplementary Fig. 11). A shift from Lhcr to Lhcf in the second belt could have EET implications, as the Lhcf subfamily lacks the Lhcr-specific chlorophyll-binding motifs and possesses a significantly shorter helix C (see below, Supplementary Fig. 10a–j).

## Gene losses enable changes in the 1st belt subfamily and architecture

Although the first belt shows an Lhcr majority across the red lineage, first-belt positions FCP1/3 show conspicuous exceptions in FCP subfamily and orientation, with consequences on EET networks and belt architecture (Fig. 2, Supplementary Fig. 11). We posit that FCP1/3's subfamily switches and rotations can be parsimoniously explained by gene losses that remove contingent PSI:FCP interactions and allow the "empty" positions to be explored and filled in new ways, as discussed below[11,37].

*M. pyrifera*'s FCP1 belongs to the RedCAP family of the LHC superfamily, as is the case for the other studied phyla of the red lineage, except for dinoflagellates, which have lost the *RedCAP* gene and filled the FCP1 position with an Lhcf protein[20,38]. RedCAPs show a family-specific extension in the B-C loop that fits into a crevasse formed by PsaB, PsaI and PsaL, forming the most extensive FCP:PSI interactions across red-lineage antennae (Fig. 2a-d). Additionally, RedCAP forms the weakest EET pathway to PSI among FCP families,

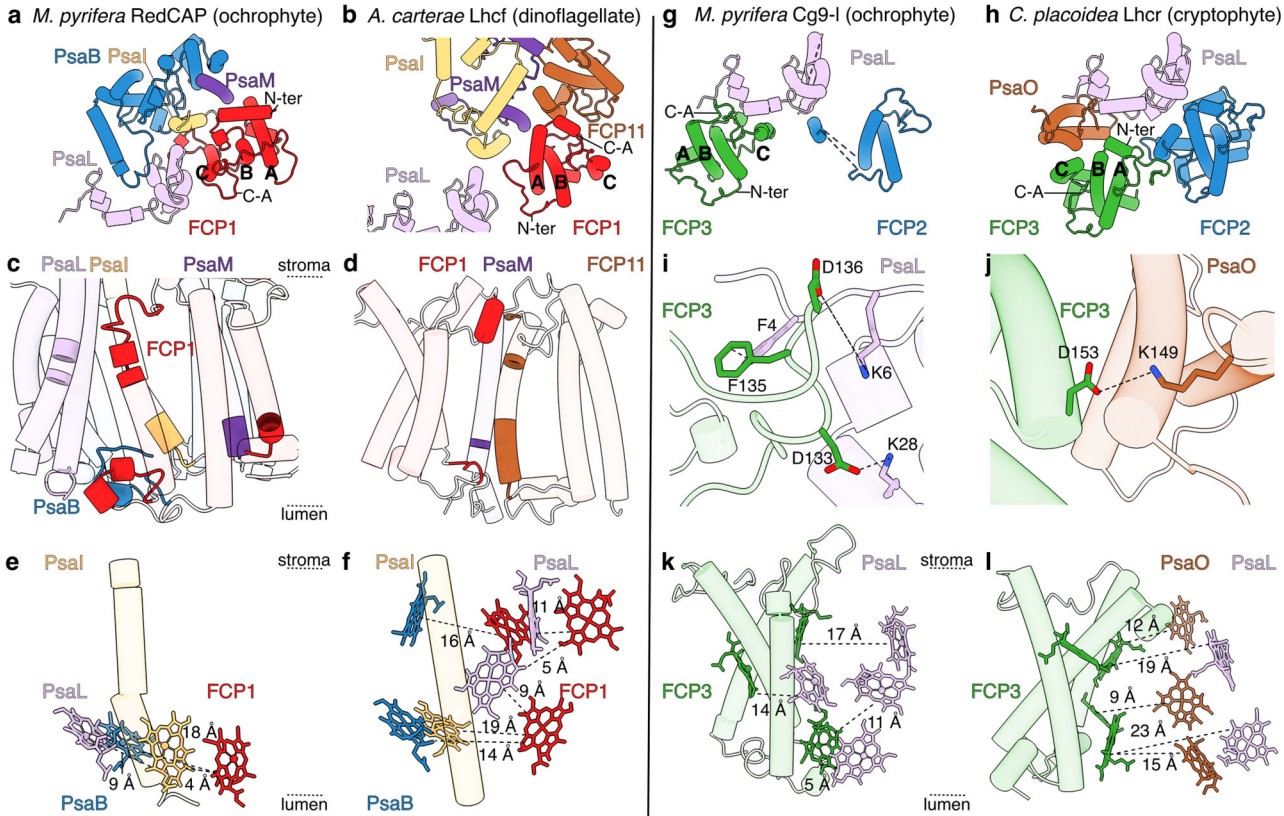

**Fig. 2 | FCP subfamily switches for FCP1 and FCP3. a–f** FCP1 switch from RedCAP to Lhcf subfamily. **g–l** FCP3 switch from Lhcr to CgLhr9-like (Cg9-l) subfamily. **a, c, e, g, i, k** Details for *M. pyrifera* (brown alga, ochrophyte, PDB: 9YGV, this work). **b, d, f** Details for *Amphidinium cartera* (dinoflagellate, PDB: 8JW0)[19]. **h, j, l** Details for *Croomonas placoidea* (cyroptophyte, PDB: 7Y7B)[15]. **a, b, g,** and **h** Relevant subunits discussed in the text, viewed from the stroma, shown in a cartoon, coloured by subunit. Helices A, B and C, C-A loop (C-A) and N-terminal loop (N-ter) labeled. Details of key protein:protein interactions viewed from the membrane (**c, d**) or stroma (**i, j**). **e, f, k, l** Interactions of key chlorophyll molecules between the partners in (**i, j**) viewed from the membrane. Edge-to-edge distances between the porphyrin rings are shown. Approximate placement of stroma and lumen is shown with dashed lines.

with only one pair of chlorophyll molecules closer than 15 Å (Fig. 2e, f). RedCAP shows a strong conservation in the FCP1 position despite its weak EET connectivity and the presence of multiple FCP subfamilies with additional chromophores in ochrophytes, haptophytes, and dinoflagellates[29] (Supplementary Fig. 11). This suggests that the FCP1 conservation is driven by the binding surface rather than by RedCAP's functional significance for light absorption or EET (Fig. 2e, f). In contrast, dinoflagellates show a loss of the *RedCAP* gene and a concomitant replacement of the FCP1 position with a pre-existing Lhcf protein[29]. Rather than the limited EET interactions of RedCAP-FCP1 (Fig. 2e), Lhcf-FCP1 contains four chlorophyll molecules that have been measured to be key EET positions from FCP1 to PsaB and PsaL[20] (Fig. 2f). Additionally, the dinoflagellate Lhcf-FCP1 is rotated ~180° relative to the "standard" orientation, using helices A and B rather than helix C to interface with PSI[19,20] (Fig. 2a, b, d). This decreases the difference in transmembrane helix height (a.k.a, hydrophobic mismatch) between FCP1 and its PSI partners from ~14 Å to ~4 Å (Supplementary Fig. 13f, g, Supplementary Data 2).

A similar situation is seen with FCP3, which switched from Lhcr in cryptophytes to Cg9-l in *M. pyrifera*, diatoms (ochrophytes), haptophytes and dinoflagellates (Supplementary Fig. 11). Thus, *M. pyrifera*'s FCP3 lacks the Lhcr protein-binding and ligand motif otherwise seen in the first belt (Supplementary Fig. 13a–c). Moreover, *M. pyrifera*'s FCP3 shows a ~135° rotation along its vertical axis relative to other first-belt FCPs and to cryptophyte FCP3 (Fig. 2g, h). This rotation brings Cg9-l closer to PSI subunit PsaL, allowing more extensive protein:protein interactions and shorter chlorophyll distances[15,16] (Fig. 2g-l). The

rotation also mitigates the hydrophobic mismatch between FCP3 and PSI by reducing the ~13 Å difference to a ~6 Å (Supplementary Fig. 13h, i, Supplementary Data 2). The changes in FCP3 family and orientation can be explained by changes in its ancestral interaction partner PsaO, which is encoded in the nucleus of red algae and cryptophytes and lost in the rest of the lineage[17,19–23,39,40]. The presence of PsaO correlates with an Lhcr FCP3 in standard orientation (red algae, cryptophytes); gene loss of *PsaO* perfectly correlates with a Cg9-l FCP3 in rotated orientation (ochrophytes, haptophytes, dinoflagellates). This strongly suggests that loss of PsaO produced a physical gap in PSI and released contingent interactions between PSI and the Lhcr FCP3, allowing for the replacement by a different family in a non-standard orientation, with re-configured EET connections (Fig. 2k, l). Moreover, the pattern of similarities across the different clades has implications for the disambiguation of evolutionary models of the red lineage (Supplementary Note 2, Supplementary Fig. 14).

A similar pattern of gene loss and LHC rotation is seen in recent structures of *Euglena gracilis*, a secondary endosymbiont of the green lineage (PDB: 9VJS)[41,42]. In this alga, LHCs in positions analogous to FCP1/9/10/11 display 180° rotations relative to available green algae structures and to the red lineage[1]. Rotation of FCP1-equivalent is associated with loss of PsaL/I. Rotation of FCP9-equivalent is correlated to the loss of PsaR and consequent rearrangements of neighbouring FCP8/10/11. In the cases where the available structures allow for inter-organismal comparisons, i.e., FCP10/11-equivalents, the rotations decrease hydrophobic mismatch with both PSI and the second-belt LHCIs (Supplementary Fig. 13j, k).

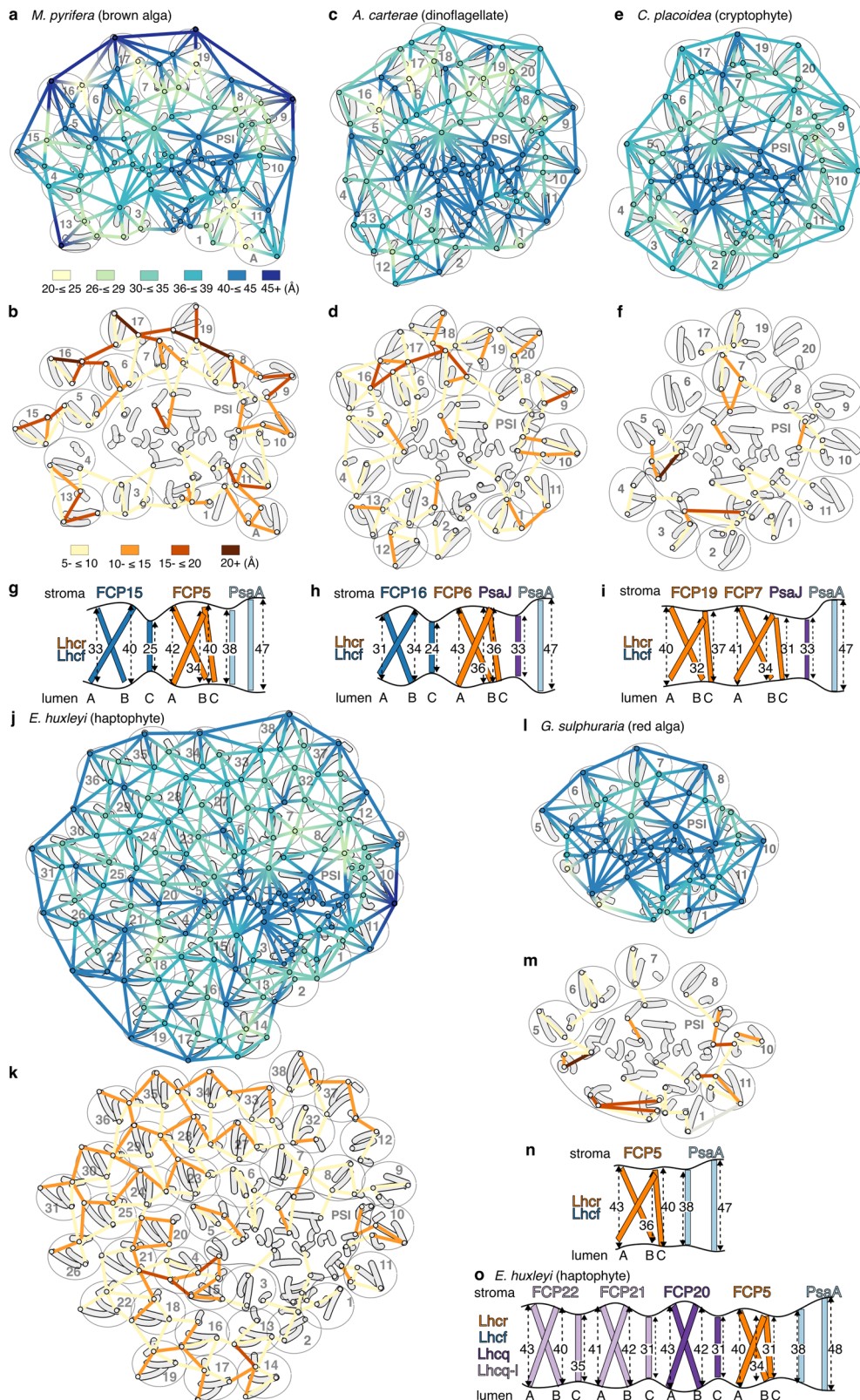

## Differences in FCP composition and orientation lead to membrane rippling

Prompted by the above rotations, we investigated transmembrane helix heights across the red-lineage antennae. Our analyses revealed that FCP orientation and subfamily identity lead to a "membrane rippling" effect across PSI-FCP in *M. pyrifera* and other red-lineage organisms (Fig. 3, Supplementary Data 2).

FCP transmembrane helices A, B, and C have characteristic amino acid lengths and membrane heights in each subfamily, ranging ~26-41 Å, consistent across red lineage phyla (Supplementary Fig. 10f–j,

**Fig. 3 | Membrane thickness differences in PSI-FCP across the red algal lineage.** Analysis for key red-lineage organisms: **a, b, g** *M. pyrifera* (PDB: 9YGV, this work); **c, d, h** *A. carterae* (PDB: 8JW0)[19]; **e, f, i** *C. placoidea* (PDB: PDB: 7Y7B)[15]; **j, k, o** *E. huxleyi* (PDB: 9JJ8)[18]; **l, m, n** *G. sulphuraria* (PDB: 9KC5)[13]. **a, c, e, j, l** Graph of membrane thickness, as measured by the height of transmembrane helices of PSI-FCP of red-lineage organisms, overlaid on helices (grey). Nodes are placed on the stromal end of each transmembrane helix, colour coded by helix height in steps of 5 Å from 20 Å (yellow) to 45+ Å (dark blue). See scale centred below (a). Edges between neighbouring helices are coloured as a gradient between the colours of the nodes. Ovals and numbers label the approximate space of the underlying subunit per nomenclature in Supplementary Fig. 12, except for *E. huxleyi* which retains its own nomenclature[18]. For *M. pyrifera*, FCP2 and FCPB are absent due to the inability to determine the FCP subfamily. **b, d, f, k, m** Graph of the difference in membrane thickness of neighbouring transmembrane helices of PSI-FCP. Nodes are placed as above colour-coded by helix-height difference in steps of 5 Å from 5 Å (yellow) to 20+ Å (maroon). See scale centred below (**b**). Differences below 5 Å are not shown. **g–i, n, o** Schematic membrane view from core to outer-belt. FCP representations are coloured by their subfamilies (Lhcr in orange, Lhcf in blue, Lhq in purple, Lhcq-like in lilac). Helix heights are denoted by arrows and labelled in Å. FCP transmembrane helices are labeled A-C. Full data provided in Supplementary Data 2.

Supplementary Data 2). In contrast, the PSI transmembrane helices show heights of ~32-48 Å, with the longest helices in the vicinity of the reaction center and decreasing heights towards the antenna. This creates a difference in membrane thickness of up to 17 Å from the center of PSI to the first belt (Fig. 3, Supplementary Data 2). Additionally, the presence of different FCP subfamilies and the orientation of the FCP relative to PSI and to other FCPs create particular patterns of membrane rippling (Fig. 3g–i, n, o). Across the lineage, Lhcr proteins arrange themselves to present helices B and C to the core, driven by FCP:FCP aromatic-residue-mediated interactions and the conserved interactions with PSI subunits (Fig. 3g–i, n, o, Supplementary Fig. 13a–c). In contrast, in the second and outer belts, Lhcq, Lhcq-like and Lhcf subunits tend to orient their short helix C towards generally longer helices A/B of the previous belt(s). In red algae and cryptophytes, which only contain Lhcr FCPs in their antennae, minimal rippling is observed (Fig. 3e, f, i, l–n). Additionally, cryptophytes' second belt Lhcrs show the same orientation as those in the first belt, suggesting that the orientation is not driven by FCP-PSI interactions, but by Lhcr properties (Supplementary Fig. 11). Haptophytes' rippling is more extensive, originating mostly from orientation effects of Lhcq and Lhcq-like FCPs throughout the outer belts (Fig. 3 j, k, o). In contrast, the outer-belt rippling in *M. pyrifera* and dinoflagellates is mostly due to their Lhcf majority, which possess the shortest helix C across FCP subfamilies. (Fig. 3a–d, g, h).

We posit two main functional implications for the transmembrane differences in the antenna. First, given that both the FCP1/3 rotations in red endosymbionts and the LHCI rotations in green endosymbiont *E. gracilis*[41,42] decrease the mismatch between neighbours, we propose that hydrophobic-mismatch-minimisation is a general and parsimonious explanation for FCP/LHC rotations across photosynthetic antennae (Supplementary Fig. 13jf–k). Second, membrane thickness affects the membrane's dielectric properties and impacts the absorption spectra of chromophores[43–45]. From this, we infer that FCP subfamilies with different transmembrane thickness could have different absorption properties even with equivalent chromophore composition. To our knowledge, this has not been directly tested. We hypothesise that the thickness of the local membrane will affect FCP absorption and EET properties and thus could provide additional selective advantages for particular FCP subfamilies at specific antenna locations. Biophysical experiments with reconstituted PSI supercomplexes, among others, will shed light on these hypotheses.

### Different predicted EET pathways in Chrysista versus Diatomista ochrophytes

The phylum Ochrophyta splits into two major clades, Chrysista and Diatomista, with vast molecular, morphological and metabolic diversity[46,47] (Supplementary Fig. 1e). Whereas diatoms (Diatomista) are the single largest contributors to marine net primary production[2,48], kelp forests (Chrysista) are the most net-primary productive per unit area[4]. Determining the differences between the photosynthetic complexes of these two ochrophyte clades will provide initial steps towards understanding kelp's high productivity. Moreover, given ochrophytes' variable positions in different red-lineage evolutionary models, understanding which structural features are brown alga- or diatom-specific versus ochrophyte-wide may help differentiate the feasibility of competing phylogenomic models[10,49,50] (Supplementary Note 2, Supplementary Fig. 14). Thus, we compared the PSI-FCP structures of *M. pyrifera* and diatoms *Chaetoceros gracilis* and *Thalassiosira pseudonana*. We focused our comparisons on antenna chromophore composition and chlorophyll distances, critical factors for EET efficiency[21–23,36] (Fig. 4, Supplementary Table 3-4). In diatoms, the strongest EET entry points into PSI have been determined to be the FCP3/4/14, FCP10/11 and FCP7/8 pathways[21–24]. We observed chromophore losses/gains and consequent EET re-organization for *M. pyrifera* at these key diatom locations (Fig. 4, Supplementary Fig. 15).

In the FCP3/4/13 region, *M. pyrifera* showed loss of eight chlorophyll molecules in FCP3/4 relative to diatoms, including chlorophyll pair FCP4 *c*420-*c*421 (Fig. 4a–e, Supplementary Fig. 15a, b). These losses increase the chlorophyll distance between FCP3/4/13 and PsaA, thus decreasing EET efficiency. This implies that FCP3-4 are weaker EET entry points into PSI in *M. pyrifera* relative to diatoms. However, given that the FCP3/4 chlorophyll complement of the other red-lineage organisms resembles *M. pyrifera*, the extensive FCP3/4/13 EET pathway is likely a Diatomista gain. Thus, this Chrysista/Diatomista difference is unlikely to influence kelp's high productivity[13–20].

*M. pyrifera* showed two lumenal chlorophyll losses in the FCP10/11 pathway, increasing EET distances between FCP10/11, as well as between FCP11/PsaB (Fig. 4a, f, g, Supplementary Fig. 15c–e). In contrast, a chlorophyll stromal gain in FCP11 placed an additional chlorophyll (chl *a*423) ~6 Å away from a modelled chlorophyll *a* pair in FCPA (chl *a*403/*a*406), one of the distince *M. pyrifera* antenna positions not seen in other organisms (Fig. 4f, g, Supplementary Fig. 15e). This pair is expected in Lhcf subunits, as seen in the model for FCP17. However, we note that the FCPA map was of medium resolution (~4 Å) and therefore these chlorophyll molecules are modelled with intermediate confidence (Supplementary Fig. 15e). Chlorophyll pairs are critical components of EET networks thanks to their energy-sink and photoprotectant properties[35,36]. The proximity between these *M. pyrifera* chlorophyll molecules, as well as with a photoprotective fucoxanthin molecule, suggests that this is an additional brown algal EET pathway from the second belt into PSI[18] (Fig. 4f, g, Supplementary Fig. 15e). Moreover, a second FCP10/11 EET pathway into PSI exists on *M. pyrifera*'s stromal side via FCP11's Lhcr-specific chl *c*415 and FCP10's chl *a*407. Their ~10 Å distance suggests they are significant players in FCP10/11 transfer into the core[35,36]. Replacement of FCP11 with a non-Lhcr FCP subfamily would increase the closest stromal distance with FCP10 to ~23 Å, suggesting pressure to conserve Lhcr in this position (Supplementary Fig. 13a–e). Overall, these differences imply a stronger predominance of the stromal transfer around FCP10/11 in *M. pyrifera*. This is likely correlated to the appearance of FCPA/B and additional FCPs in brown algal relative to other red-lineage organisms, and could contribute to functional differences (Supplementary Fig. 8).

In *M. pyrifera*'s FCP7/8 region, larger stromal distances due to chlorophyll losses as well as a lumenal reconfiguration between FCP8 and PsaR (a.k.a Psa28) imply a stronger lumenal FCP7/8 entryway into PSI relative to diatoms[23] (Fig. 4h–l, Supplementary Fig. 15f–i). *M.*

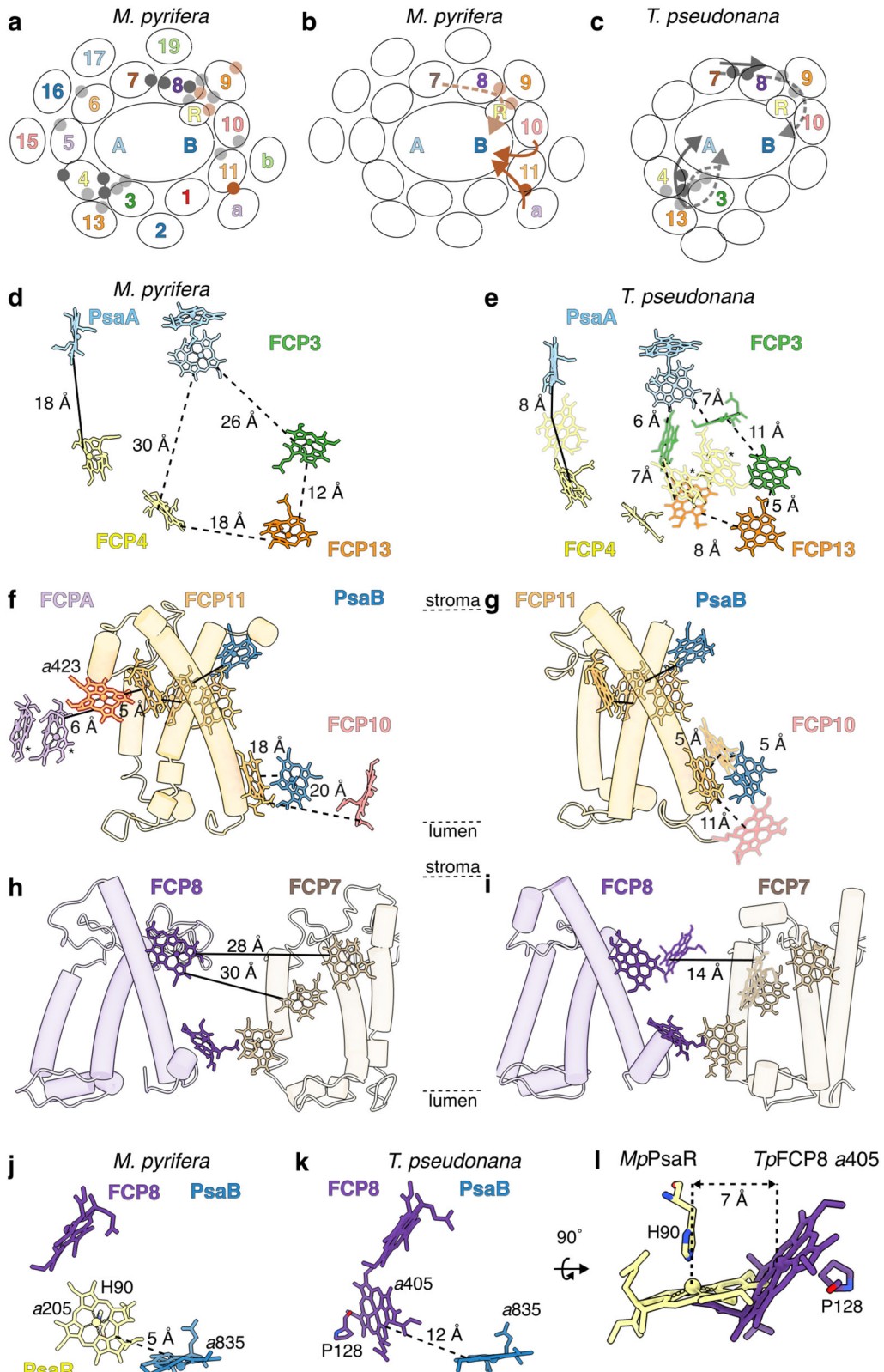

*pyrifera*'s FCP8 lumenal side has lost diatom FCP8 chl *a*405 (coordinated by diatom FCP8-Pro128). However, this loss is compensated by *Mp*PsaR's chl *a*205 coordinated by PsaR-His90 (Fig. 4j–l). This shift of chlorophyll coordination from FCP8-Pro to PsaR-His changes coordination type, brings chl *a*205 ~ 7 Å closer to PsaB chl *a*835 (Fig. 4j–l), and places the chlorophyll in a more hydrophobic protein environment. The protein motif that houses MpPsaR-His90 is conserved in the

Phaeophyceae and various sister clades within Chrysista but is missing in Diatomista ochrophytes and other red-lineage phyla, suggesting it was acquired in the Chrysista clade (Supplementary Fig. 16). Given that coordination type and environment affect chlorophyll absorption, emission and excited-state lifetimes[51–53], we predict significant differences in the light absorption and EET in the FCP7/8-PsaR region of PSI-FCP in Chrysista versus Diatomista. Furthermore, this and other

**Fig. 4 | Summary of key differences in PSI-FCP chromophore arrangement between *M. pyrifera* and diatoms. a–c** Schematic summaries of chromophore arrangements in *M. pyrifera* (PDB: 9YGV, this work) (**b**) and diatom *Thalassiosira pseudonana* (PDB: 8ZEH)[24] (**c**). Dark grey, dark brown indicate chromophores on the stromal side; light grey, light brown, on the lumenal side. Subunits coloured as in Fig. 1 and labeled without Psa/FCP prefixes for clarity. a, FCPA; b, FCPB. **a** Summary of chromophores lost (grey) or gained (brown) in *M. pyrifera* with respect to diatoms. Key EET pathways in *M. pyrifera* (**b**) and diatom *T. pseudonana* (**c**). Stromal pathways in solid arrows; lumenal pathways in dashed arrows. Participating chromophores are indicated with circles (dark, stromal; light, lumenal). Key chlorophyll differences between *M. pyrifera* (**d, f, h, j**) and *T. pseudonana* (**e, g, i, l**) are shown for various regions. Chromophores gained in *M. pyrifera* are marked with a thick brown

border in *M. pyrifera* panels. Chromophores lost in *M. pyrifera* are marked with a grey border in *T. pseudonana* panels. Edge-to-edge distances between porphyrin rings are indicated with solid lines for stromal connections and with dashed lines for lumenal connections. Asterisks denote chlorophyll pairs. Detailed chlorophyll nomenclature and distances in Supplementary Fig. 15. **d, e** Differences in FCP3/4/13-PsaA. **f, g** Differences in FCP11/A-PsaB. Note that FCPA is absent in the available diatom structures. **h, i** Differences in FCP7/8. **j–l** Chlorophyll migration from FCP8 (*T. pseudonana* chl *a*405) to PsaR (*M. pyrifera* chl *a*205). **l** Superposition of *T. pseudonana* (*Tp*) FCP8-chl *a*312 and *M. pyrifera* (*Mp*) chl *a*205, aligned by PsaR. Coordinating residues and approximate Mg-Mg distance between the chlorophyll molecules are shown.

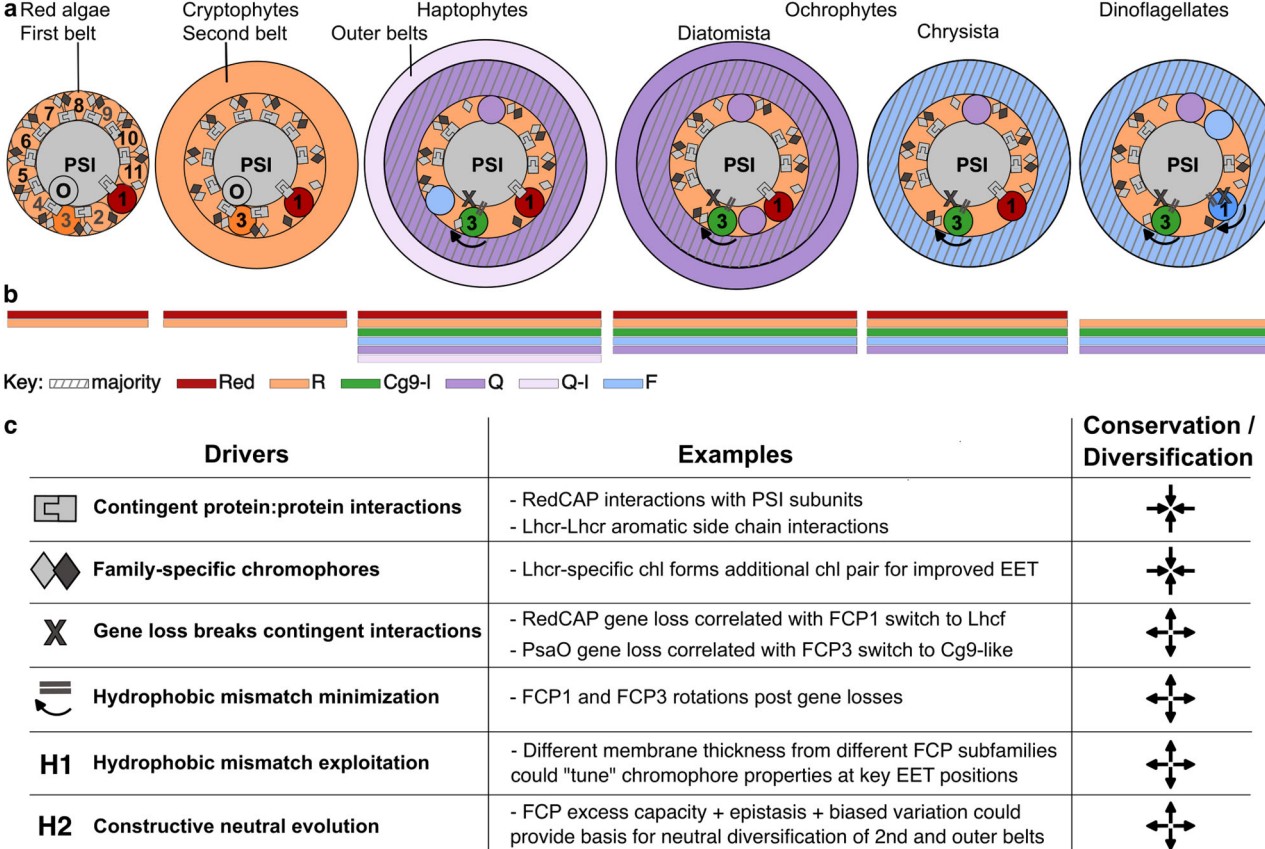

**Fig. 5 | Drivers of antenna conservation and diversification in the red lineage. a** Schematic representation of PSI-FCP supercomplexes across the red lineage, based on currently available structures discussed in text. FCP subfamilies represented indicated by colours as in (**b**). Symbols for drivers of antenna evolution overlaid on PSI-FCP as in (**c**). For simplicity, belts are represented as full annuli

rather than partial belts. **b** Schematic representation of FCP subfamilies present in each clade. **c** Summary of antenna evolution drivers and examples discussed in text. Converging arrows indicate driver for conservation; diverging arrows indicate a driver for diversification. H1, H2, hypothetical drivers proposed in the text.

differences in the lumenal region of FCP8 across the red lineage suggest that the FCP8 lumen is a hotspot for antenna diversification across the red lineage, which could underlie photosynthetic differences.

## Discussion

Here we present the structure of brown alga *M. pyrifera*'s PSI-FCP supercomplex, showing a two-belt FCP antenna arranged differently from currently studied red-lineage organisms, including other ochrophytes (Figs. 1–4, Supplementary Fig. 11). Our comparison of the antenna arrangement across the two major ochrophyte clades showed Chrysista-specific and Diatomista-specific features that may be implicated in photosynthetic differences (Fig. 4). Spectroscopic studies are needed to confirm whether brown algal PSI has more prominent lumenal EET pathways than diatom PSI and determine the relative

contributions of the FCP8-PsaR and FCP13/3/4-PsaA entry pathways into PSI.

From our comparisons of structures and FCP subfamily composition, we conclude that several drivers can explain the conservation and diversification patterns of red-lineage antennae (Fig. 5). Contingent protein:protein interactions between PSI and FCPs, together with Lhcr-specific protein and EET features, maintain an Lhcr majority in the first belt across the lineage. Gene loss, likely from failed transfer to the nucleus upon higher-order endosymbiosis (Supplementary Note 2), releases contingent constraints and allows the lineage to explore different FCP:PSI and FCP:FCP interactions (e.g., FCP1/3 subfamily switches). For these interactions, hydrophobic mismatch minimisation leads to FCP rotations that may establish distinct EET connections. Additionally, we speculate that the FCP subfamilies'

hydrophobic thickness differences may provide additional "tuning" of chromophore properties that may be further exploited at critical positions in the EET network. The fact that endosymbiosis-driven gene loss is also correlated with antenna-protein rotation and hydrophobic mismatch minimization in green secondary endosymbiont *E. gracilis* supports our notion that these are general drivers shaping antenna architecture in complex chloroplasts beyond the red lineage[41,42].

Across red-derived organisms, the subfamily composition of the second and outer belts is much more diversified than that of the first belt. This diversification accompanies the expansion of the FCP family into various new subfamilies after the red algal engulfment[29,30]. A key open question is whether this diversification is driven by selective advantages or by neutral mechanisms. The outer-belt diversification has been typically explained as an adaptation to clade-specific environments[6,13,29,30]. It is clear that the presence of different accessory chromophores in the antenna proteins leads to differences in the absorption spectra of PSI-antenna supercomplexes that can allow for better light absorption at different depths[54]. However, it is not evident whether the differential incorporation of LHC subfamilies at specific antenna locations leads to improved light absorption, photosynthetic function and fitness ultimate selection. This is particularly unclear for red-lineage organisms that share depths in the epipelagic zone yet have diverse antenna (e.g., haptophyte *I. galbana* vs. diatom *C. gracilis* vs. cryptophyte *C. placoidea*)[55,56]. Conversely, some organisms share antenna subfamily composition yet live at very different depths and habitats (e.g., haptophyte *I. galbana* vs haptophyte *Emiliania huxleyi*)[55,57,58]. Therefore, we speculate that antenna diversification may not be purely adaptive and that neutral evolutionary processes may also be in play. We propose that the theory of constructive neutral evolution (CNE), which posits that complexity can arise neutrally, provides an alternative framework to understand antenna diversification[11,37].

Key components of the CNE framework are excess capacity, epistasis and biased variation[11,37]. In the case of the PSI antenna, FCP duplication and expansion from Lhcr into Lhcf, Lhcq, Lhcx, Lhcz and Cg9-l subfamilies provide excess capacity[29]. Many of the new genes might contain slightly deleterious changes, e.g., one fewer chromophore in the stroma as is the case for Lhcf *vs* Lhcr (biased variation). The excess FCP capacity would lead to many gratuitous interactions between FCPs while assembling the outer belts, e.g., incorporating an Lhcf rather than Lhcr protein in the FCP17 position. The suboptimal mutation of the Lhcf might be rendered neutral by epistasis, e.g., if small mutations in its Lhcr partner result in a favourable binding interaction. Then, this new composition might drift to fixation, effectively "locking in" the originally gratuitous interaction, even if the original change (incorporating an Lhcf) was slightly deleterious on its own. Thus, the interdependency leads to an increase in complexity that arose neutrally rather than by positive selection[11,37].

The PSI core is a highly efficient photochemical machine achieving ~99% quantum efficiency, surpassing PSII (~85%)[35,36]. This high efficiency is thought to underlie the antenna's expansion, accommodating increased travel times from the outer belts without too-detrimental a compromise to PSI's quantum efficiency[35,36]. For instance, the haptophyte *E. huxleyi* achieves 95% efficiency with its eight-belt FCP antenna[18]. In terms of CNE, the high PSI efficiency would provide a permissive landscape to incorporate the FCP excess capacity, leading to larger, diversified, more complex antennae. Thus, we hypothesise that the diversification of PSI's outer antenna results, at least in part, from neutral processes enabled by the permissivity of PSI's high efficiency, especially at positions that do not play critical EET roles. This remains to be experimentally tested.

This first cryoEM structure of a brown algal complex lays the groundwork to understand kelp's high photosynthetic productivity "from the bottom up", to support the development of kelp forests as blue-carbon ecosystems[26]. Further structural, biochemical, biophysical and genetic experiments are needed to test our hypotheses and derive the full complement of drivers shaping PSI-FCP architecture, composition and function.

## Methods

### Chloroplast purification

Fresh *Macrocystis pyrifera* blades were obtained from Monterey Bay Seaweeds. Their chloroplasts were isolated based on previous protocols, with modifications[59]. For each isolation preparation, ~1 kg of fresh fronds was rinsed with deionized water and treated with $H_2O$-HCl (pH 6.0) for 30 min to remove surface-adsorbed materials. The tissue was then rinsed, finely chopped and incubated twice in a 0.1 M sodium citrate buffer (pH 8.0) at 4 °C under gentle stirring to remove alginates[60]. Samples were centrifuged at 5,000 x g for 10 min and washed with deionized water to remove residual citrate. The pellet was resuspended in ~1:8 g:mL of homogenization buffer (50 mM Tris, 150 mM NaCl, 10 mM phosphate, 1 mM EGTA, 0.35 M mannitol, 1% w:v PVP-40, 0.1% w:v BSA, 8 mM cysteine and boric acid at pH 8.0). To further decrease the viscosity of the resuspension and aid in organelle isolation, recombinant alginate lyase (Alg2A from *Flavobacterium* sp. S20, 6 mg $L^{-1}$)[61] was added, and the mixture was incubated for 1 h at 4 °C. The tissue was then homogenised using a 4-L blender (Waring, Stamford, CT, USA, model: CB15BU) at high speed for three consecutive 5-second cycles, filtered through two layers of Miracloth (MilliporeSigma, Burlington, MA, USA) and centrifuged at 5,000 x g for 45 min at 4 °C. The chloroplast pellet, typically yielding ~250 g $kg^{-1}$ of initial tissue, was washed in a buffer containing 50 mM HEPES, 0.3 M mannitol, and 1 mM EDTA (pH 7.5), and further purified by layering it on top of a 2-step Percoll gradient (40% v:v over 80% v:v) (Sigma-Aldrich, St. Louis, MO, USA) diluted in a 20 mM HEPES buffer at containing 0.33 M sucrose at pH 7.5. The gradient was centrifuged at 3000 x *g* for 20 min at 4 °C. The chloroplast-enriched fraction was collected manually, washed twice with a buffer containing 20 mM HEPES, 1 mM EDTA, 50 mM NaCl, and 10% glycerol, adjusted to pH 7.5 with NaOH, by centrifuging at 4000 x *g* for 5 min. The pellets were then aliquoted in volumes of 2 mL, and flash-frozen in liquid nitrogen and stored at −70 °C.

### Chloroplast membrane wash

To isolate thylakoid membranes, the purified chloroplasts were subjected to a hypotonic shock by resuspension in chilled Milli-Q water at a ratio of 10 mL:1g water:membranes and homogenised with 100 strokes using a glass homogeniser with a tight pestle. Subsequently, 3 M KCl was added to adjust the suspension to a final concentration of 150 mM KCl and re-homogenised with an additional 100 strokes. Membranes were collected by centrifugation at 45,000 x g. The pellet (membrane fraction) was solubilised in buffer (20 mM Tris-HCl, 150 mM NaCl, 1 mM EDTA, 10% v:v glycerol, 10 U $mL^{-1}$ DNase I, 2 mM DTT and 0.002% PMSF, at pH 7.4) and homogenised with another 100 strokes using a glass homogeniser and tight pestle. After final homogenization and centrifugation at 45,000 x g for 30 min at 4 °C, the membrane fraction was obtained as a brown pellet enriched in thylakoid membranes. The chlorophyll concentration of the sample was determined spectrophotometrically following the equations of Jeffrey and Humphrey[62]. To extract the chlorophyll, samples were incubated with 0.1% Triton X-100 (Thomas Scientific, Swedesboro, NJ, USA) for 10 min at room temperature, followed by dilution in 90% acetone and incubation with gentle tumbling for 10 minutes at room temperature. Samples were spun at 14,000 x *g* for 15 minutes and the supernatant was collected. The absorption of the supernatant was measured 350–750 nm at 2 nm intervals, using a Spectramax M2 spectrophotometer (Molecular Devices, San Jose, CA, USA). Membranes were aliquoted in a volume of 1 mL at 0.7 mg $mL^{-1}$ of chlorophyll content, flash-frozen in liquid nitrogen and stored at

−70 °C in a buffer containing 20 mM Tris, 50 mM NaCl, 1 mM EDTA, 2 mM DTT, 0.002% PMSF and 30% glycerol at pH 7.4.

## PSI-FCP supercomplex extraction and purification

To extract PSI-FCP complexes, the washed membranes were solubilised in 1% w:v digitonin (MilliporeSigma, Burlington, MA, USA) (detergent-to-chl w:w ratio of 73:1, or 12:1 for dataset in Supplementary Fig. 8) using extraction buffer (15 mM HEPES, 20 mM KCl, and 5 mM NaCl at pH 7.5) for 1 h at 4 °C under gentle tumbling. The solubilised material was clarified by centrifugation at 16,000 x $g$ for 15 min, and the supernatant was concentrated to 100–200 µL using an Amicon Ultra Centrifugal Filter, 100 kDa MWCO (MilliporeSigma, Burlington, MA, USA). Linear sucrose gradients (20–50% w:v) were prepared in extraction buffer (15 mM HEPES pH 7.5, 20 mM KCl, and 5 mM NaCl) containing 0.01% GDN (Anatrace, Maumee, OH, USA). The concentrated sample was layered onto the gradients and ultracentrifuged at 263,600 x $g$ for 22 h at 4 °C. Gradients were fractionated using a Gradient Station (Biocomp Instruments, Fredericton, NB, Canada). Fractions with a maximum absorbance peak at 676–678 nm were collected and selected for cryoEM sample preparation below.

## HPLC pigment analysis

Pigments were extracted from *M. pyrifera* PSI-FCP using acetone with 0.1% Triton X-100 v:v. Pigment analysis was performed by high-performance liquid chromatography (HPLC) on an Agilent 1100 Series system equipped with a diode array detector and an Agilent Eclipse XDB column (5 µm, 4.6 ×150 mm). Separations were carried out at 20 °C with a flow rate of 1 mL min-1 using solvent A (MeOH/H2O = 90:10 [v:v]) and solvent B (ethyl acetate). The gradient was programmed as follows: 0–20 min, 0-100% B; 20–22 min, 100% B; 22–23 min, 100–0% B; and 23-28 min, 0% B. Pigments were detected at 445 nm with fill spectra collected from 300-800 nm, and identified by comparison to authentic standards of chlorophyll a, c, β-carotene, fucoxanthin, violaxanthin (Sigma-Aldrich, St. Louis, MO, USA) and zeaxanthin (AmBeed, Buffalo Grove, IL, USA). Quantitative standard curves were prepared in 100% acetone using the following concentrations: 0.5 µg mL-1, 1 µg mL-1, 5 µg mL-1, 10 µg mL-1, 50 µg mL-1, and 100 µg mL-1. Absolute quantification of PSI pigments was achieved by measuring peak areas. Carotenoid standards contained isomeric mixtures and peak areas were summed for quantification.

## FCP molecular phylogeny reconstruction

The full set of FCP genes in *Macrocystis pyrifera* was identified from the Macpyr2 (CI_03 v1.0) genome resource[63] in PhycoCosm[64] by performing BLASTP searches using FCP sequences from *C. gracilis*[65] and *Ectocarpus siliculosus*. Transcriptomic data from *M. pyrifera*[63] were also searched, revealing additional sequences not annotated in the genome. This yielded 57 non-redundant FCP sequences across the genome and transcriptome. For subsequent phylogenetic analysis, sequences that were incomplete, truncated or showed ambiguous annotation were removed. For the phylogenetic analysis, we included FCP/LHC sequences previously identified and validated from *E. siliculosus*, *C. gracilis* and *T. pseudonana*[65]. To root the tree, we used a curated set of canonical green-algal LHCs from *Volvox africanus* (UniProt ID: A0A8J4EUW6), *Volvox reticuliferus* (UniProt ID: A0A8J4LST4), *Chlorella vulgaris* (UniProt ID: A0A9D4TGL7) and *Chlamydomonas reinhardtii* (UniProt ID: A8IKC8). All sequences were aligned with MAFFT on the EBI web server using the L-INS-i strategy, BLOSUM62, a gap extension/offset penalty set to 0.123 and a maximum of 100 iterative refinement cycles. Misaligned or compositionally biased regions were corrected by manual inspection in Aliview v1.30[66]. The alignment file was used to reconstruct phylogenetic trees. Maximum-likelihood (ML) phylogenies were inferred with PhyML 3.0[67] under the best-fit amino-acid substitution model selected by Smart Model Selection using AIC. Branch support was quantified with the SH-like approximate likelihood ratio test (SH-aLRT) and the approximate Bayes test (aBayes) as implemented in PhyML 3.0[67]. Branch lengths are reported as substitutions per site. Resulting topologies were visualised and minimally edited to adjust tip label orientation and improve overall readability in FigTree v1.4.4[68].

## CryoEM grid preparation and data collection

The sample used for cryoEM grid preparation consisted of the above sucrose-gradient-purified PSI-FCP supercomplex from *M. pyrifera* at a final chlorophyll concentration of 1.0 mg mL⁻¹. Selected fractions from the purification above were buffer-exchanged into a buffer containing 15 mM HEPES (pH 7.5), 20 mM KCl, 5 mM NaCl, 0.01% (w:v) GDN (Anatrace, Maumee, OH, USA), and 0.1% (w:v) digitonin (MilliporeSigma, Burlington, MA, USA). CryoEM grids (Quantifoil 1.2/1.3 300 mesh Gold, GmbH, Germany) were glow-discharged for 30 s at 30 mA (PELCO easiGlow, Ted Pella Inc, Redding, CA, USA), followed by a 10 s hold prior to sample application. Grids were prepared by a double blotting procedure. First, 2 µL of sample was applied, held for 30 s and manually blotted for 5 s outside the plunge freezer. Subsequently, 4 µL of sample was added, incubated for 10 s, and blotted for 8 s under controlled conditions (18 °C and 90% relative humidity) in a Leica EM GP2 cryo-plunger (Leica, Wetzlar, Germany). Grids were then plunge-frozen in liquid ethane using the same instrument. CryoEM data were collected using a 300 kV microscope equipped with a Falcon 4i detector operated in electron counting (EC) mode at SLAC National Accelerator Laboratory, Stanford, CA, USA. Data acquisition was performed using EPU software. A total dose of 50 e⁻ Å⁻² was fractionated into 40 frames over an exposure time of 4.95 s, resulting in a dose rate of 1.25 e⁻ Å⁻² frame⁻¹. Movies were collected at a physical pixel size of 1.217 Å, corresponding to a magnification yielding 1.27 Å pixel⁻¹ on the detector. A total of 15,690 micrographs were acquired, with a fluence of 6.82 e⁻ pix⁻¹ (10.10 e⁻ Å⁻² s⁻¹).

## CryoEM image processing and composite map generation

Motion correction and contrast transfer function (CTF) estimation were performed using CTFFIND4.1[69] in Relion v4.0.1[70]. Particle picking was initially guided by manual picking and subsequently used to train a Topaz model[71]. Particles were transferred to cryoSPARC v4.4.1[72] for successive rounds of 2-dimensional classifications and ab-initio volume generation before heterogeneous refinement and re-extraction of a curated particle set at the original pixel size. Non-uniform refinement was performed, followed by core-focused Local Refinement using a soft binary mask around PSI generated in UCSF ChimeraX v1.10.1[73]. For FCP1/3/7/8/10, individual soft masks were used for local refinement of the corresponding FCP. FCP2/4/5/6/9/11 required an additional masked classification step to separate particles in which the FCP density was not strongly present in the particle set used for the above refinements, followed by local refinements focused on each FCP. Refinement of the second-belt FCPs required a hierarchical alignment approach. For each second-belt FCP, particle stacks were first aligned using the neighbouring first-belt subunit as a reference, followed by masked classification and local refinement around each second-belt subunit. Maps originating from the local refinement of the PSI and of the individual FCPs were aligned to the overall map of the PSI-FCP supercomplex in ChimeraX and integrated into a composite map in Phenix[74].

## Model building and protein sequence assignment

Sequences of *M. pyrifera*'s PSI subunits were identified by BLAST searches using the annotated subunits of *Ectocarpus siliculosus* (Genome ID: CAID01000000) as queries against the Macpyr2 genome[63] (*Macrocystis pyrifera* CI_03 v1.0) in the Phycosm database[64]. The top hits selected for each subunit were used to generate initial models with AlphaFold 3[75]. These structures were rigid-body fit into cryoEM density maps in ChimeraX and manually real-space-refined in Coot 0.9.8.96[76].

For PsaR, which was not initially detected by BLAST, its density was carved and subjected to ModelAngelo[60] to obtain a putative sequence assignment, which was then used as a BLASTP query against the NCBI ClusteredNR dataset in GenBank, yielding a *M. pyrifera* homologue as the top hit, which was modelled and validated as above.

To begin the assignment of the protein identities of the FCP densities in our maps, we initially searched for the top hits of each *C. gracilis* FCP sequence in the *M. pyrifera* genome in PhycoCosm[63] using BLASTP searches. The three top-scoring *M. pyrifera* hits for each diatom FCP query were retained as candidates. Structural predictions for each top hit, including seven chlorophyll molecules to stabilise the fold, were generated with AlphaFold3, rigid-body fit in ChimeraX[73] and manually refined in Coot. To determine the protein identity among the three top candidates, the candidate sequences were aligned, and the positions showing the highest amino-acid structural variability were noted. The fit of the predicted model of each candidate to the map was manually inspected and validated. For FCP7, initial searches with the diatom and *Ectocarpus* queries yielded unsatisfactory candidates (clear mismatch with map). Thus, we undertook a ModelAngelo-based approach and used the predicted sequence as a query in a tBLASTn search against *M. pyrifera* transcriptomic database in Phycosm[63]. The best hit corresponded to an unannotated transcript, from which the open reading frame was translated into a protein sequence, modelled, inspected and validated. The second-belt FCPs whose resolution was not sufficient to assign a protein sequence (FCP15/16/19/A) were modelled as a poly-alanine chain, based on the FCP17 model.

Ligand assignment was guided by homologous diatom PSI-FCP supercomplex structures and refined by visual inspection. Chlorophyll molecules were modelled, where porphyrin-like densities were clearly resolved. Thus, unassigned subunits (FCP2/15/16/19/A/B) lack models for all carotenoids and most of the expected chlorophylls. Throughout, chl *a* and chl *c* were distinguished based on the continuity of phytol chain density and based on previous assignment in the highest-resolution diatom structure available. The atoms of the chl *a* phytol chain were removed where the density did not allow for atomic modelling.

The full model was real-space refined against the composite map in Phenix[74], and validated with MolProbity[77].

### FCP subfamily assignment

To assign subfamilies to *M. pyrifera* FCPs with an assigned protein sequence (FCP1/3-11/13/17), subfamilies were determined based on the location of the sequence in *M. pyrifera*'s phylogenetic tree. For the *M. pyrifera* FCPs without assigned sequences (FCP2/15/16/19/A/B, poly-alanine models), we used map-model correlation scores to identify the best-matching family. Models of each of the assigned families were fit into the focused-refined maps of each unassigned FCP, using ChimeraX command fitmap, with manual adjustment as needed. The models used to represent the subfamilies were *M. pyrifera*'s FCP4-11 for Lhcr, FCP8 for Lhcq, FCP1 for RedCAP, FCP3 for CgLhcr9-like, FCP17 for Lhcf. To determine the quality of map-to-model fit, Q-scores were determined in ChimeraX. Maps of decreasing resolution were produced with the volume tools job in cryoSPARC by changing the low-pass filter parameter.

### Transmembrane helix height measurements

Transmembrane helices were measured using the Coot distance tool on the available PSI structures. To account for the curvature of the helices, a minimum of three scaling measurements were taken, scaled relative to the field of view of the membrane in Inkscape, and then averaged for the reported membrane lengths. For PsaA/B at least three transmembrane helices were measured for each organism and averaged. Subunits with RMSD scores <0.55 Å per ChimeraX were considered the same and reported as the same measurement. To determine FCP subfamily helix length averages, a minimum of five FCP

subfamily representatives from each available PSI organism were measured and averaged to achieve a total average for each subfamily across the red lineage.

### Reporting summary

Further information on research design is available in the Nature Portfolio Reporting Summary linked to this article.

## Data availability

The atomic coordinates for the *M. pyrifera* PSI-FCP supercomplex have been deposited (PDB: 9YGV) [https://doi.org/10.2210/pdb9YGV/pdb]. Maps corresponding to the composite map (EMDB: 76146) as well as 19 focused-refinement maps for each component of the composite map, have been deposited [https://www.ebi.ac.uk/emdb/EMD-76146]. Details are available in Supplementary Table 2. The raw cryoEM data have been deposited (EMPIAR:12998) [https://doi.org/10.6019/EMPIAR-12998].

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

## Acknowledgements

We thank Monterey Bay Seaweeds for providing kelp samples for our research, and S. Nuzhdin for providing access to the *M. pyrifera* genome prior to publication. We thank D. Barty, K. Bejar, Y. Xiang, H. Tiet, H. Wimboeck, C. Wang and M. Xu for kelp organelle isolations, C. Richards for lab management, F. Guo, I. Fries and A. Cassago for technical cryoEM support. We thank members of the Lagarias and Letts labs, especially M. Ayala-Hernandez, for advice, shared materials and equipment. We thank J. del Mármol, M. Iwai, J. Letts, N. Rockwell, S. Theg and members of the Letts lab for comments on the manuscript. Some of this work was performed at the Stanford-SLAC Cryo-EM Center (S2C2), which is supported by the National Institute of General Medical Sciences (1R24GM154186). The content is solely the responsibility of the authors and does not necessarily represent the official views of the National Institutes of Health.

## Author contributions

J.W., P.M., H.M.O., R.R., G.W., P.Z., M.M. developed methodology, performed experiments and analysed data; J.W., P.M., H.M.O., G.W., V.G.T.D., P.Z., M.M. produced figures; M.M. acquired funding and administered the project; M.M. and P.Z. supervised research; M.M., J.W., P.M. wrote the manuscript; all authors reviewed and edited the manuscript.

## Funding

This work was supported by start-up funds from the University of California, Davis Departments of Plant Biology, College of Biological Sciences and Center for the Advancement of Multicultural Perspectives (CAMPOS), as well as by a Boomerang Carbon Capture Research Award (MM). MM is a University of California, Davis Hellman Society Fellow and CAMPOS Scholar.

## Competing interests

The authors declare no competing interests.
