## [Transparent Peer Review file · Nature Communications]

Structure of giant kelp Photosystem I-FCP uncovers drivers of antenna evolution across the red lineage

Corresponding Author: Dr María Maldonado

Version 0:

Reviewer comments:

Reviewer #1

(Remarks to the Author)

This manuscript reports a cryo-EM structural analysis of a PSI-FCP supercomplex from kelp, representing a previously uncharacterised photosynthetic clade. The technical execution is sound, and the quality of the structural data appears to be high. From a methodological standpoint, the work is carefully performed. The authors successfully determine the overall architecture of the PSI-FCP complex and provide a detailed description of subunit organisation, antenna composition and pigment arrangement. The study is generally well presented, and the structural models appear internally consistent. As such, the manuscript constitutes a solid piece of structural work and provides a useful addition to the existing body of comparative data on photosynthetic supercomplexes, which will be of interest to specialists in algal photosynthesis and antenna organisation.

Assessment of novelty and conceptual advance

Determination of a PSI-FCP structure from a new clade does constitute a degree of novelty. However, in the current landscape of photosynthetic structural biology, structural determination in a new lineage alone is no longer sufficient to meet the criteria of a journal at this level. The manuscript largely follows a descriptive framework that has become common in recent studies of photosynthetic complexes. The emphasis is placed on determination of overall architecture, assignment of pigment positions, description of subunit arrangement, and interspecies structural comparison. While these aspects are competently executed, they remain primarily descriptive in nature. Resolution is repeatedly highlighted, yet the manuscript does not clearly demonstrate how the reported resolution leads to conceptual insights that extend beyond detailed structural description. Although the high resolution enables careful measurements of helix length, pigment distance and membrane thickness, these observations are not developed into broadly applicable principles or mechanistic frameworks. More importantly, while the manuscript raises multiple interesting themes—including antenna diversification, excitation-energy-transfer reorganization, membrane thickness variation, and red-lineage evolution—it does not converge on a clearly defined central biological or physical question. As a result, the conceptual advance appears diffuse, and the study does not sufficiently address the implicit “so what?” question expected at this level.

Comparative emphasis and limited brown algal-centred insight

Although the authors explicitly recognise that diatoms and brown algae are evolutionarily distant lineages, the manuscript relies almost entirely on diatoms as the interpretative reference for the brown algal PSI-FCP structure. Consequently, the study is heavily comparative in its emphasis, with much of the discussion driven by contrasts with diatom PSI-FCP complexes. While such comparisons are undoubtedly informative, they dominate the narrative to the point that the biological significance of the brown algal system itself is insufficiently developed. Many of the reported features are framed primarily as departures from diatom organisation, rather than being examined as intrinsic properties of brown algae. As a result, it is difficult for the reader to identify which aspects of the observed PSI-FCP architecture can reasonably be regarded as genuinely characteristic of brown algae, as opposed to reflecting broader lineage-level differences revealed through inter-lineage comparison. This imbalance weakens the conceptual focus of the manuscript. Despite presenting the first PSI-FCP structure from brown algae, the study falls short of articulating a clear biological message centred on brown algal photosynthesis. Instead, brown algae function largely as a comparative reference used to reassess diatom-derived generalisations, rather than as the primary biological subject in their own right.

Pigment assignment and EET

In peripheral FCP regions, the local resolution is clearly limited. Under such conditions, direct discrimination between

chlorophylls and carotenoids based solely on electron density is intrinsically challenging. Nevertheless, the criteria used for pigment assignment, as well as the associated uncertainty, are not sufficiently discussed. As a consequence, model-dependent pigment assignments are presented in a relatively definitive manner despite the limitations of the data. This issue is non-trivial, as pigment assignment underpins subsequent interpretations related to excitation energy transfer, functional implications and evolutionary arguments. Greater transparency regarding uncertainty, alternative assignments and confidence levels would therefore be essential. Given these uncertainties, interpretations related to excitation energy transfer should be treated with particular caution. In this study, EET pathways are discussed extensively and used to infer functional and evolutionary implications; however, because EET analyses critically depend on accurate pigment identity, position and orientation, the ambiguity in pigment assignment undermines the robustness of such conclusions. Under these circumstances, EET-related interpretations would be more appropriately presented in a simplified or subsidiary manner rather than as a central element of the manuscript. Emphasising structural organisation and comparative architecture, while clearly delimiting EET analyses as tentative or hypothesis-generating, would lead to a more balanced and methodologically sound presentation.

Use of Constructive Neutral Evolution (CNE)

Constructive Neutral Evolution is a well-established evolutionary framework and is not introduced here as a new concept. In this manuscript, however, CNE is invoked extensively in the Discussion without being supported by dedicated analyses, quantitative tests or figures that would allow the hypothesis to be evaluated. As presented, CNE functions less as a testable hypothesis and more as an explanatory narrative that compensates for the absence of direct mechanistic or functional evidence. While such speculation may be appropriate as a conceptual perspective, it should be clearly framed as such and not positioned as a substantive conclusion. Without comparative or quantitative evolutionary analyses, the invocation of CNE remains conjectural.

Figure presentation and conceptual synthesis

A further limitation of the manuscript is the absence of a unifying summary figure that encapsulates the central conclusions of the study. While Figures 4 and 5 provide detailed structural comparisons, they primarily illustrate local differences in pigment arrangement and subunit organisation rather than conveying an overarching conceptual framework. As a result, the reader is left without a clear visual synthesis of how the reported structural observations collectively advance understanding of antenna evolution, excitation-energy-transfer reorganisation, or red-lineage diversification. For a manuscript positioned at the level of Nature Communications, a schematic or integrative figure that distils the main conceptual message would be essential.

Assignment and nomenclature of small PSI subunits

From the perspective of a non-specialist reader, the assignment and nomenclature of the PSI subunit referred to as PsaR raise a number of unresolved questions. In earlier work, including a 2008 study describing the purification and characterisation of *Acaryochloris* PSI, a small PSI-associated component was designated as Psa27. The introduction of a numerical designation at that time could reasonably be interpreted as reflecting a historical progression beyond the established letter-based naming scheme for PSI subunits. Against this background, it is not immediately clear why more recent structural studies have introduced or reintroduced letter-based designations, such as PsaR, for small PSI-associated components. For a reader without detailed prior knowledge of PSI subunit nomenclature, it is difficult to ascertain whether Psa27 and PsaR are intended to represent the same evolutionary or structural entity, lineage-specific variants of a related component, or fundamentally different classification concepts based on distinct criteria. This uncertainty is further compounded by the coexistence of multiple nomenclature schemes in the recent literature. While some studies, including an eLife report published in 2024, provide a historical and evolutionary framework for PSI subunits, other recent structural analyses adopt alternative assignments without clearly reconciling them with earlier designations. As a result, the continuity of PSI subunit nomenclature across the literature is not readily apparent to the reader. Clarification of the rationale underlying the present assignment—specifically, how it relates to previously defined components such as Psa27, and why a particular nomenclature scheme is preferred—would therefore be highly beneficial. An explicit discussion of the criteria used for subunit identification, including sequence homology, structural correspondence, genomic context, and evolutionary conservation, would greatly improve transparency and facilitate comparison across studies.

Scope and grounding of the concluding discussion

In the concluding part of the Discussion, the manuscript broadens its interpretation to encompass wider ecological and societal contexts, including the high photosynthetic productivity of kelp and the role of kelp forests as blue carbon ecosystems. While these themes are undoubtedly of considerable interest, their connection to the structural observations reported in this study is not sufficiently developed. In particular, it remains unclear how the structural features identified in the brown algal PSI-FCP complex—such as antenna architecture, FCP composition, or the proposed excitation energy transfer (EET) pathways—translate into demonstrable advantages in photosynthetic performance at either the organismal or ecosystem level. In the absence of direct physiological, biochemical, or ecological evidence, these broader interpretations appear largely aspirational rather than being firmly supported by the data presented. Consequently, the final section of the Discussion risks extending the implications of the structural analysis beyond what can be robustly justified, moving from well-supported comparative observations towards more speculative statements concerning productivity and carbon sequestration. A more cautious framing, or a clearer distinction between conclusions directly supported by the data and longer-term perspectives, would substantially enhance the coherence and credibility of the Discussion.

Overall evaluation

This work undoubtedly provides valuable comparative structural data and useful information for the specialist photosynthesis community. However, it does not establish a new general principle, a substantial conceptual advance, or a clear mechanistic or functional breakthrough. In its current form, the manuscript therefore appears better suited to a more

specialised journal focused on structural or comparative photosynthesis research, rather than to a venue seeking broad conceptual impact.

Reviewer #2

(Remarks to the Author)

This is an interesting cryo-electron microscopy (cryo-EM) study of the photosystem I complex from the giant kelp *Macrocystis pyrifera*. The data identify differences in the chlorophyll network of the PSI-FCP supercomplex, as well as variations in the thickness of the transmembrane hydrophobic layer across the supercomplex, suggesting potential functional consequences for photochemistry. This work lays an important foundation for understanding the high photosynthetic productivity of kelp, reveals new factors contributing to the conservation and diversification of antenna systems, and provides insights into the evolutionary relationships among red-lineage organisms.

Overall, this is an excellent study performed by a top-tier group with strong expertise. However, it is not clearly described how the FCP proteins were identified and matched to specific sequences. Is the identification unambiguous, or are there remaining uncertainties?

Major comments:

1. The basis for identifying the FCP proteins is not clearly demonstrated, making it difficult to evaluate the assignments. In particular, the map quality of the second belt is insufficient, and the evidence supporting their assignment as Lhcf proteins is weak.
2. Although differences in membrane thickness are observed, the relationship between these differences and functional implications remains unclear.
3. The FSC curve in Extended Data Fig. 2 reaches the Nyquist limit, suggesting that the resolution estimation may not be accurate. The data were collected using a Falcon 4i detector, and it should be possible to import the data with up-sampling. Doing so might further improve the resolution and potentially enhance the density quality of the peripheral (second-belt) FCPs.

Minor comments:

Extended Data Figures and Supplemental Figures are mixed, which may cause confusion and should be unified.

Reviewer #4

(Remarks to the Author)

Manuscript review

I have enjoyed reviewing this manuscript by Weissman and Maturana from Maria Maldonado's lab. Overall the manuscript is well written and clearly explain its points and the figures are informative and visually appealing.

The study presents the first high-resolution structure of the PSI-FCP supercomplex from *M. pyrifera*, revealing a two-belt antenna organization distinct from diatoms and other red-lineage algae. The work provides detailed insights into FCP subfamily composition, chromophore arrangement, and potential functional implications for excitation energy transfer. The authors are also proposing constructive neutral evolution as a mechanism driving outer-belt diversification. The methodology is sound, analyses are appropriate and conclusions are well supported by the data. This study represents a significant advance in structural and evolutionary understanding of photosynthetic antennae, with relevance for photosynthesis research, evolutionary biology, and marine ecology. *M. pyrifera* forms dense kelp forests that are ecological powerhouses, fueling coastal food webs, shaping marine habitats and driving primary production on a massive scale. Clarifying the structure and function of giant kelp's photosynthetic machinery is therefore key to understanding the productivity and resilience of these vital ecosystems.

Minor editorial revisions (such as ensuring consistent figure numbering, correcting grammar and adding a few clarifications) would further improve readability, but these do not affect the overall quality or impact of the work.

Not clear sentences or mistakes:

- In the main text, the authors refer to "Supplementary Fig. X", whereas in the figure legend of the Supplementary file the figures are labeled as "Supplemental Fig. X". Please use consistent terminology throughout the manuscript.

Page2:

Rephrase please:

11 Whereas photosystems are structurally and functionally conserved, antenna systems and the chromophores
12 they contain are highly diverse, tuned to optimal absorption of wavelengths in different

13 environments (6).

Page3:

Rephrase please:

32 To understand drivers of conservation and diversification the antenna and shed light on
33 functional and evolutionary relationships between red lineage phyla,

Page4:

Repetition & grammar:

3 The Lhcr majority in the first belt of the *M. pyrifera* PSI antenna is seen in
4 seen across all the red lineage, and it is an absolute majority in red algae and cryptophytes ...

Repetition & grammar:

16 center (Extended data Fig. 7n,o). Indeed, fast transfer (<10 ps) between between these...

Rephrase please:

22 ...tree, the resolution of

23 FCP15/16/19/A was insufficient resolution to assign protein sequences.

Grammar:

40 ...switch and rotation can be parsimoniously explained by them resulting from of gene losses that

41 broke contingent...

Page5:

Repeat:

18 This decreases the difference in transmembrane helix height between between...

Page6:

Rephrase please:

33 Mismatch minimization also provides a

34 parsimonious explanation as to why FCP rotations are favored belt upon gene loss and subfamily

35 switches (Fig. 2).

Page8:

Rephrase please:

14 Clear support for the Pietluch model comes from FCP9/13, where cryptophytes and haptophytes

15 are closely aligned, but different from ochrophytes and dinoflagellates, themselves closely

16 aligned (10) (Fig. 1c).

Questions and suggestions:

General observations:

- Several "Extended Data Fig. X" figures are either not cited (Extended Data Fig. 1) in the main text or are referenced in a non-sequential order (e.g., Extended Data Fig. 2-5 are the first to be mentioned). Please ensure that all Extended Data figures are cited in the appropriate sections of the text and follow a logical numbering order.

Page3:

13 "After optimizing protocols to decrease the viscosity of the kelp sample, we isolated a

14 chloroplast-enriched fraction from fresh *M. pyrifera* blades, from which we extracted and

15 purified PSI-FCP for structural determination using cryogenic electron microscopy (cryoEM)

16 (Extended Data Figs. 2-5, Supplementary table 1-3)." —> Please mention the detergent used in this purification, as it is one of the differences from previously published structures.

37 "Note that the FCP number refers to the position in the antenna, not to the gene

38 name. For clarity, all antenna proteins are referred to as "FCP" using our numbering, even if the

39 LHC subunit binds other chromophores in certain organisms. —> It would be helpful to include a table that links each

FCP discussed in this manuscript to the corresponding FCPs in other organisms, facilitating comparison with existing

nomenclature. This table would support what shown in Supplementary figure 4

Page4:

1 "Similarities in sequence, subunit composition and architecture of the supercomplex are strongest

2 in PSI, weakening with increasing distance from the core." —>I would recommend mentioning from the beginning whether the PSI core is conserved across brown algae.

14 "This Lhcr-specific chlorophyll is ~6 . away from chl 407 of the adjacent FCP, posing

15 these chlorophyll molecules for efficient excitation energy transfer (EET) into PSI's reaction

16 center (Extended data Fig. 7n,o)". —> The authors state that a ~6 Å distance is conducive to efficient excitation energy

transfer. It would be helpful to specify the distance range typically required for efficient EET, to better support this interpretation.

28 “The subfamilies of FCPB, as well as FCP2 in the first belt, remained unassigned.” —> It would be helpful to clarify in which organisms FCPB proteins have previously been assigned to specific subfamilies, if any. This would help place the present limitation in a broader comparative context.

Materials and Methods

- How was assessed the purity and quality of the sample?
- Which magnification was used for data collection?

Looking at the maps:

- Extended Data Fig.2

Looking at Extended Data Fig. 2, it would be helpful to show the distribution of particles that contributed to the final full map. Do the particles exhibit a preferred orientation? Including a particle distribution plot for the final map, as well as a local resolution map, would greatly help in assessing the quality of the map.

- Extended Data Fig. 3-4-5

Some FCPs in the first belt and all FCPs in the second belt are observed only in subsets of particles. Could the authors comment on whether this reflects the purification conditions (light/dark), the detergent used, or possibly the functional dynamics of the supercomplex, such as transient assembly under particular physiological states?

Version 1:

Reviewer comments:

Reviewer #1

(Remarks to the Author)

I am broadly satisfied with the authors' responses and revisions. Overall, the response has been scientifically serious and appropriate. In particular, it is to the authors' credit that they did not retreat into a defensive posture, but instead acknowledged uncertainty where necessary and carefully reconsidered the framing, emphasis, and scope of their claims. These revisions have clearly improved the scientific rigour and internal coherence of the manuscript.

More broadly, however, I remain concerned that, in part of the recent cryo-EM literature in photosynthesis, functional and evolutionary implications, as well as perceived manuscript impact, are sometimes foregrounded before the scientific robustness of the structural analysis itself and the validity of the associated interpretations have been subjected to sufficiently critical scrutiny. The consequence is that the boundary between conclusions directly supported by the data and those that remain, at least for the time being, hypothetical can become unnecessarily blurred. Against that background, the authors' willingness in the present case to recalibrate the scope of their claims, make uncertainty explicit, and undertake substantial revision rather than press overstated conclusions is clearly to be commended.

This preference for scientific restraint over rhetorical inflation distinguishes the revised manuscript positively. It is a strength not only of the present study, but also an attitude that would serve the field well more generally.

On that basis, I view the revised manuscript positively. Quite apart from this specific review process, I hope the authors will continue to pursue their future work with the same degree of scientific seriousness, rigour, and restraint.

Reviewer #2

(Remarks to the Author)

The author's comments are convincing. I have no more comments.

Reviewer #4

(Remarks to the Author)

After the first round of review, the article “Structure of giant kelp Photosystem I-FCP uncovers drivers of antenna evolution across the red lineage” by Weissman et al. has become significantly clearer and more complete. The purpose and relevance of the work are now better articulated and the addition of Figure 5 effectively illustrates the broader significance of this study for kelp algae.

The authors have addressed my comments, as well as those of the other reviewers, in a thorough and constructive manner. The revisions, corrections and clarifications have been well integrated into the manuscript, improving its overall quality.

Looking ahead, it would be interesting to investigate these complexes in situ to better understand how these similar yet distinct photosynthetic machineries are organized under native conditions across different organisms.

I thank the authors again for their careful responses and recommend this article for publication.

Best regards,
Bianca Introini

Response to reviewers

Reviewer #1 (Remarks to the Author)

This manuscript reports a cryo-EM structural analysis of a PSI–FCP supercomplex from kelp, representing a previously uncharacterised photosynthetic clade. The technical execution is sound, and the quality of the structural data appears to be high. From a methodological standpoint, the work is carefully performed. The authors successfully determine the overall architecture of the PSI–FCP complex and provide a detailed description of subunit organisation, antenna composition and pigment arrangement. The study is generally well presented, and the structural models appear internally consistent. As such, the manuscript constitutes a solid piece of structural work and provides a useful addition to the existing body of comparative data on photosynthetic supercomplexes, which will be of interest to specialists in algal photosynthesis and antenna organisation.

Thank you for your words and your thoughtful and thorough reading of our manuscript.

We fully agree with the reviewer on multiple points, most of which originated from our lack of clarity and explicitness in an effort to reduce words in the original submission. We have thoroughly considered the reviewer's comments and made the relevant edits, as described below.

Assessment of novelty and conceptual advance

Determination of a PSI–FCP structure from a new clade does constitute a degree of novelty. However, in the current landscape of photosynthetic structural biology, structural determination in a new lineage alone is no longer sufficient to meet the criteria of a journal at this level. The manuscript largely follows a descriptive framework that has become common in recent studies of photosynthetic complexes. The emphasis is placed on determination of overall architecture, assignment of pigment positions, description of subunit arrangement, and interspecies structural comparison. While these aspects are competently executed, they remain primarily descriptive in nature. Resolution is repeatedly highlighted, yet the manuscript does not clearly demonstrate how the reported resolution leads to conceptual insights that extend beyond detailed structural description. Although the high resolution enables careful measurements of helix length, pigment distance and membrane thickness, these observations are not developed into broadly applicable principles or mechanistic frameworks. More importantly, while the manuscript raises multiple interesting themes—including antenna diversification, excitation-energy-transfer reorganization, membrane thickness variation, and red-lineage evolution—it does not converge on a clearly defined central biological or physical question. As a result, the conceptual advance appears diffuse, and the study does not sufficiently address the implicit “so what?” question expected at this level.

We appreciate the thoughtful way in which you have pointed out that our original manuscript lacked focus and convergence and that our advances were not shining through. Addressing this concern through some editing, figure re-organization and the creation of a summary figure has significantly improved our revised manuscript.

We fully agree that resolution and structures in and of themselves have limited utility and that structures should be the starting points for the generation of biological insights and hypotheses. This is our strong stance in the laboratory and the spirit in which we wrote our manuscript. This is why we did not emphasize nominal resolution in the manuscript, mentioning resolution mostly in the context of it not being sufficient to make certain interpretations. We reviewed and edited all sections to ensure no undue mentions of resolution were present.

In our original manuscript, we strived to go beyond structural descriptions to derive insight and hypotheses informed by evolutionary and biophysical principles and synthesize conceptual advances that were not previously appreciated in other PSI-antenna papers that remain heavily descriptive. This novel conceptual synthesis is now explicit in our new conceptual-summary figure, which encapsulates our conceptual contributions (Fig. 5). The figure displays four identified drivers and two additional speculative drivers, all of which incorporate evolutionary and biophysical principles. These drivers will provide the field with new metrics and vocabulary (central questions) with which to analyze and test structures and their insights. They also provide clear hypotheses to explicitly test with biophysical, genetic and evolutionary experiments and observations. Together, these drivers allow the field to determine the “so what?” of current and future structures beyond mere description going forward.

During the time of revision, a structure of the green secondary endosymbiont *Euglena gracilis* PSI-antenna became available (Li et al, 2026). Encouragingly, this structure also shows gene losses correlated with LHC rotations and hydrophobic minimization (added to the text and Supplementary Fig. 13). This suggests that the drivers we have identified are general to higher-order endosymbiosis and the architecture of photosystem antenna in complex chloroplasts, beyond the red lineage.

Fig. 5. Drivers of antenna conservation and diversification in the red lineage. **a** Schematic representation of PSI-FCP supercomplexes across the red lineage, based on currently available structures discussed in text. FCP subfamilies represented indicated by colours as in (b). Symbols for drivers of antenna evolution overlaid on PSI-FCP as in (c). For simplicity, belts are represented as full annuli rather than partial belts. **b** Schematic representation of FCP subfamilies present in each clade. **c** Summary of antenna evolution drivers and examples discussed in text. Converging arrows indicate driver for conservation; diverging arrows indicate driver for diversification. H1, H2, hypothetical drivers proposed in text.

Reference:

Li, K., Qin, BY., Zhang, YZ. et al. Structure and energy transfer of a far-red-absorbing euglenophyte PSI–LhcE–LhcbM supercomplex. *Nat Commun* (2026). <https://doi.org/10.1038/s41467-026-70067-1>

Comparative emphasis and limited brown algal-centred insight

Although the authors explicitly recognise that diatoms and brown algae are evolutionarily distant lineages, the manuscript relies almost entirely on diatoms as the interpretative reference for the brown algal PSI–FCP structure. Consequently, the study is heavily comparative in its emphasis, with much of the discussion driven by contrasts with diatom PSI–FCP complexes. While such comparisons are undoubtedly informative, they dominate the narrative to the point that the biological significance of the brown algal system itself is insufficiently developed. Many of the reported features are framed primarily as departures from diatom organisation, rather than being examined as intrinsic properties of brown algae. As a result, it is difficult for the reader to identify which aspects of the observed PSI–FCP architecture can reasonably be regarded as genuinely characteristic of brown algae, as opposed to reflecting broader lineage-level differences revealed through inter-lineage comparison. This imbalance weakens the conceptual focus of the manuscript. Despite presenting the first PSI–FCP structure from brown algae, the study falls short of articulating a clear biological message centred on brown algal photosynthesis. Instead, brown algae function largely as a comparative reference used to reassess diatom-derived generalisations, rather than as the primary biological subject in their own right.

We sincerely appreciate your comment that the salience of kelp in its own right was not standing out. We carefully reviewed the whole manuscript, and in particular the EET section, to improve the kelp centeredness. We also appreciate the point that our original manuscript read too heavily focused on diatoms. Our intention was to avoid presenting diatoms as central organisms as the standard to which brown algae are compared. Rather, we intended to point out that ochrophytes are a highly diverse class and that diatoms should not be taken as the “model” organism/clade, as is frequently presented in many publications. However, due to the ease of genetic resources available and the consequently large number of studies available on diatoms, sometimes the comparison is unavoidable even when one intends to make a more general point across the red lineage. We have reviewed our writing in the sections for structure/phylogeny, drivers of conservation/ diversification and endosymbiotic implication to make our stance clearer.

The specific comparisons to diatoms are retained in the section on EET differences between brown alga and diatoms. In this section, we highlight diatoms while also pointing out broader red-lineage comparisons (e.g., FCP3/4/13 case). The rationale for the direct diatom comparison is mani-fold:

First, kelp and diatoms two major marine primary producers in the ocean: while kelp are the most net-productive marine ecosystem per unit area (Pessarrona et al, 2022), diatoms are the single largest producer clade (Field et al, 1998, Sun et al, 2025). Given their respective saliency, it is critical to understand the differences between these two groups (see also “Scope and grounding conclusion” section).

Second, given their ecological significance and larger availability of studies, diatoms are implicitly viewed as model or reference organisms for ochrophytes—a highly diverse clade. Given that, up to now, the only ochrophyte PSI-FCP structures have been of diatoms, many of their composition, EET and other features have been assumed or implicitly presented as representative of the whole ochrophyte clade. Through our comparative analysis of kelp, diatom and the rest of the red lineage, we have uncovered that many of those features are in fact diatom-specific (e.g., the FCP3/4/13 EET pathway), and that kelp have more similarities with the rest of the red lineage. This needs to be pointed out in our manuscript to prevent future mis-interpretations and because it is the first time that this conclusion is possible (given the availability of the brown algal structure).

Lastly, ochrophytes occupy very different positions in the three competing models of red-lineage evolution. As described in Supplementary Fig. 1, ochrophytes are thought to give rise to a) haptophytes and myzozoa (Stiller et al), b) no further red-lineage clades (Bodyl et al) or c) myzozoa only (Pietluch et al). Therefore, understanding the differences between Chrysis and Diatomista ochrophytes and determining which features are sub-class-specific and which are ochrophyte-wide is critical for the correct use of structures to help disambiguate phylogenomic models.

We recognize that these points were not clearly stated in our original manuscript and thank the reviewer for pointing it out. We have now added edited the initial and other paragraphs of EET section to this effect.

References:

Bodyl A, Stiller JW, Mackiewicz P. Chromalveolate plastids: direct descent or multiple endosymbioses? *Trends Ecol Evol.* 2009 Mar;24(3):119-21; author reply 121-2. doi: 10.1016/j.tree.2008.11.003. Epub 2009 Feb 4. PMID: 19200617.

Field CB, Behrenfeld MJ, Randerson JT, Falkowski P. Primary production of the biosphere: integrating terrestrial and oceanic components. *Science.* 1998 Jul 10;281(5374):237-40. doi: 10.1126/science.281.5374.237. PMID: 9657713.

Pessarrodona A, Assis J, Filbee-Dexter K, Burrows MT, Gattuso JP, Duarte CM, Krause-Jensen D, Moore PJ, Smale DA, Wernberg T. Global seaweed productivity. *Sci Adv.* 2022 Sep 16;8(37):eabn2465. doi: 10.1126/sciadv.abn2465. Epub 2022 Sep 14. PMID: 36103524; PMCID: PMC9473579.

Pietluch F, Mackiewicz P, Ludwig K, Gagat P. A New Model and Dating for the Evolution of Complex Plastids of Red Alga Origin. *Genome Biol Evol.* 2024 Sep 3;16(9):evae192. doi: 10.1093/gbe/evae192. PMID: 39240751; PMCID: PMC11413572.

Stiller JW, Schreiber J, Yue J, Guo H, Ding Q, Huang J. The evolution of photosynthesis in chromist algae through serial endosymbioses. *Nat Commun.* 2014 Dec 10;5:5764. doi: 10.1038/ncomms6764. PMID: 25493338; PMCID: PMC4284659.

Sun D, Jia Y, Yang S, Lang S, Ye Z, Li Z, Wang S. Global patterns in primary production of marine phytoplankton taxonomic groups. *Global and Planetary Change.* 2025; 255. <https://doi.org/10.1016/j.gloplacha.2025.105057>.

Pigment assignment and EET

In peripheral FCP regions, the local resolution is clearly limited. Under such conditions, direct discrimination between chlorophylls and carotenoids based solely on electron density is intrinsically challenging. Nevertheless, the criteria used for pigment assignment, as well as the associated uncertainty, are not sufficiently discussed. As a consequence, model-dependent pigment assignments are presented in a relatively definitive manner despite the limitations of the data. This issue is non-trivial, as pigment assignment underpins subsequent interpretations related to excitation energy transfer, functional implications and evolutionary arguments. Greater transparency regarding uncertainty, alternative assignments and confidence levels would therefore be essential. Given these uncertainties, interpretations related to excitation energy transfer should be treated with particular caution. In this study, EET pathways are discussed extensively and used to infer functional and evolutionary implications; however, because EET analyses critically depend on accurate pigment identity, position and orientation, the ambiguity in pigment assignment undermines the robustness of such conclusions. Under these circumstances, EET-

related interpretations would be more appropriately presented in a simplified or subsidiary manner rather than as a central element of the manuscript. Emphasising structural organisation and comparative architecture, while clearly delimiting EET analyses as tentative or hypothesis-generating, would lead to a more balanced and methodologically sound presentation.

Thank you for your comment. We fully agree that chromophore assignment is non-trivial, as it underpins interpretations related to EET and photosynthetic function. Given the significance of this matter, we adopted a very cautious approach for the modelling of the chromophores, as described below. Moreover, given that we did not directly measure EET in this manuscript, all our interpretations in the EET section are based on pathways that have been measured in diatoms and other red-derived organisms and have been determined to be central pathways (Xu et al 2020, Feng et al 2025). This is also why EET implications are only one aspect of our manuscript and, in our opinion, not a central element. We have added several sentences in the EET section to this effect, e.g.: *“Whether these differences are related to differences in photosynthetic efficiency remains to be experimentally tested.”*

With respect to our chromophore modeling, we would like to reassure the reviewer that, given their very different chemical structures, chlorophylls and carotenoids can be discriminated from the cryoEM density up to very low resolutions (~ 10 Å). See figure below for a comparison of densities for chlorophyll *a*, chlorophyll *c*, fucoxanthin and violaxanthin at 3 Å – 10 Å resolution using FCP8 (2.97 Å overall resolution) as a representative example. In the figure, we applied low-pass filtering to the experimental density to simulate progressively lower-resolution conditions for chlorophyll *a*, chlorophyll *c*, fucoxanthin and violaxanthin. As the figure shows, key structural features, such as the porphyrin ring of chlorophylls versus the elongated backbone of carotenoids, remain distinct up to resolutions twice as low as the lowest resolution in our focused maps.

Nevertheless, although key features of chlorophylls versus carotenoids are easily differentiable up to low resolutions, we were much more conservative and cautious in our modelling of chromophores. We differentiated our approach between assigned Psa and FCP subunits, i.e., those for which the resolution is high enough to identify protein sequence from the density (2.7 Å–3.8 Å in our structure) and unassigned FCP subunits, as described below.

As described in our Methods, for *assigned* Psa and FCP subunits, our initial chromophore modeling was based on the modeling for the highest-resolution diatom structure available (Xu et al, 2020). In the diatom PSI case, the improved maps of Xu et al (2020) vs Nagao et al (2020) allowed for the updated assignment of several chl *c* as chl *a* for *C. neogracilis* (the phytol chain of chl *a* is flexible and thus is not visible in maps with certain lower local resolutions). After initial assignment based on diatoms, we carefully reviewed our own density and made necessary changes where our resolution was sufficient to break the ambiguity. It is indeed possible that some chl *a/c* assignments will be corrected upon publication of a higher resolution structure, as seen in the diatom case. With our current high-resolution structure for the assigned subunits, we base ourselves on the homologous assignment plus manual inspection and remain open to correction as science progresses. Moreover, we removed the atoms of the chl *a* phytol chain where the density did not allow for atomic modeling, as is standard in the structural biology field.

For *unassigned* FCP subunits (FCP2/15/16/19/B, resolutions of 4.7 Å-5.2 Å), we used an even more cautious and restrained approach. We only retained chlorophylls that showed clear density for the porphyrin ring (modelled based on existing chromophores in other organisms), we did not include any carotenoids in unassigned FCPs. In this revision, we re-inspected the unassigned FCPs and removed 9 overall chromophores from FCP15/16/19/A). Note that FCP-2 and FCP-B models do not now contain any chromophores. We have now included these details in the revised Methods: “*Chlorophyll molecules were modeled where porphyrin-like densities were clearly resolved. Thus, unassigned subunits (FCP2/15/16/19/A/B) lack models for all carotenoids and most of the expected chlorophylls*”.

Importantly, the only chromophores from unassigned subunits from the second belt for which we cautiously posit EET hypotheses are chl *a*403/*a*406 in FCPA. Given the medium resolution for this subunit (3.97 Å), we transparently showed the density and the modeling for these chlorophyll molecules in Supplementary Fig. 15 (previously, Supplementary Fig. 5), so that readers can evaluate the level of confidence of our related statements and implications. To make the uncertainty explicit, in the revised version we have added the sentence: “*This pair is expected in Lhcf subunits, as seen in the model for FCP17. However, we note that the FCPA map was of medium resolution (4 Å) and therefore these chlorophyll molecules are modelled with intermediate confidence* (Supplementary Fig. 15e).”

References:

Feng Y, Li Z, Yang Y, Shen L, Li X, Liu X, Zhang X, Zhang J, Ren F, Wang Y, Liu C, Han G, Wang X, Kuang T, Shen JR, Wang W. Structures of PSI-FCPI from *Thalassiosira pseudonana* grown under high light provide evidence for convergent evolution and light-adaptive strategies in diatom FCPIs. *J Integr Plant Biol.* 2025 Apr;67(4):949-966. doi: 10.1111/jipb.13816. Epub 2024 Dec 13. PMID: 39670505.

Nagao R, Kato K, Ifuku K, Suzuki T, Kumazawa M, Uchiyama I, Kashino Y, Dohmae N, Akimoto S, Shen JR, Miyazaki N, Akita F. Structural basis for assembly and function of a diatom photosystem I-light-harvesting supercomplex. *Nat Commun.* 2020 May 18;11(1):2481. doi: 10.1038/s41467-020-16324-3. PMID: 32424145; PMCID: PMC7235021.

Xu C, Pi X, Huang Y, Han G, Chen X, Qin X, Huang G, Zhao S, Yang Y, Kuang T, Wang W, Sui SF, Shen JR. Structural basis for energy transfer in a huge diatom PSI-FCPI supercomplex. *Nat Commun.* 2020 Oct 8;11(1):5081. doi: 10.1038/s41467-020-18867-x. PMID: 33033236; PMCID: PMC7545214.

Use of Constructive Neutral Evolution (CNE)

Constructive Neutral Evolution is a well-established evolutionary framework and is not introduced here as a new concept. In this manuscript, however, CNE is invoked extensively in the Discussion without being supported by dedicated analyses, quantitative tests or figures that would allow the hypothesis to be

evaluated. As presented, CNE functions less as a testable hypothesis and more as an explanatory narrative that compensates for the absence of direct mechanistic or functional evidence. While such speculation may be appropriate as a conceptual perspective, it should be clearly framed as such and not positioned as a substantive conclusion. Without comparative or quantitative evolutionary analyses, the invocation of CNE remains conjectural.

We fully agree that CNE should not be positioned as a substantive conclusion if no experiments have been done to test any CNE-related hypotheses. In the Discussion, we invoke CNE as follows:

- “We propose that the theory of constructive neutral evolution (CNE) provides an alternative to understand this diversification.”
- “We posit [in revision: “*hypothesise*”] that constructive, neutral processes shape the antenna architecture and composition, particularly at positions that do not play critical roles in the central EET pathways.”
- “In terms of CNE, the high PSI efficiency would provide a permissive landscape to incorporate the FCP excess capacity, leading to larger, diversified, more complex antennae. Thus, the diversification of PSI’s outer antenna across the red lineage need not be positively adaptive: it may be a response to neutral processes enabled by the permissivity of PSI’s high efficiency.

Our focus was not to test CNE. Rather, we are, to our knowledge for the first time, proposing CNE as a conceptual framework on which to build testable hypotheses on antenna diversification. Our Discussion spells out how the CNE framework/ mechanism would operate for the case antenna diversification, illustrating the key components of CNE (excess capacity, epistasis, biased variation) in this context. We then integrate the CNE lens with well-established functional observations of PSI quantum efficiency to support our hypothesis that antenna diversification need not be positively adaptive (i.e., that it could be neutral).

Based on our analysis of PSI antennae across the red-derived lineage, we propose constructive *neutral* evolution as an alternative conceptual framework to understand antenna diversification. This proposal contrasts sharply with current (untested) assumptions that diversification is a *positive* adaptation to maximize light absorption in organism-specific environments. This is assumed by others even though many red-derived phototrophs have overlapping habitats (depth in the water column) in the epipelagic zone (Latasa et al, 2017, Bracher et al, 2020). Moreover, diversification of the 2nd-belt majority is evident even within each clade, e.g., Lhcr/Lhcf for *C. gracilis* and Lhcq for *T. pseudonana*. It is clear that the presence of different pigments between organisms leads to differences in the absorption spectra of PSI-FCP supercomplexes that allows for better absorbance of light at different depths, e.g., increased blue-light absorption by chl *c* and fucoxanthin (Buchel, 2019). However, it remains to be tested whether their differential incorporation to FCP of different subfamilies, in different locations significantly changes the photosynthetic function of PSI, and whether they affect organismal fitness and positive selection. In other words, what is the differential effect on photosynthetic fitness of a particular subfamily at a particular antenna location?

Thus, our intention is to open the discussion on this topic by considering that the subfamily diversification of the outer belts may not have a *positive* effect on photosynthetic outcomes; rather, the outer belt may be acquiring LHC diversity by *neutral* processes. We stress that we are *not* suggesting neutral selection to be the only or the most salient type of selection shaping the antenna, as explicitly stated in the second bullet point above. To begin to test the differential functional effects of LHC subfamilies, the absorbance properties of FCPs from different subfamilies should first be measured. Then, PSI+FCP supercomplexes could be reconstituted with varying subfamily compositions and tested for their photosynthetic efficiency,

among others. This significant biophysical undertaking would begin to show whether diversification of other belts is an adaptation with improved photosynthetic outcomes, which could be positively selected for. Other types of experiments beyond our expertise would need to address whether these are/could be selected for positively (rather than through neutral processes). We have clarified these points in the discussion. In short, we are in full agreement with the reviewer that CNE remains a conjectural, alternative framework to be tested.

References:

Bracher A, Xi H, Dinter T, Mangin A, Strass V, von Appen WJ, Wiegmann S. High Resolution Water Column Phytoplankton Composition Across the Atlantic Ocean From Ship-Towed Vertical Undulating Radiometry. *Front Mar Sci.* 2020 Apr 21;7:235. doi: 10.3389/fmars.2020.00235.

Büchel C. Light harvesting complexes in chlorophyll c-containing algae. *Biochim Biophys Acta Bioenerg.* 2020 Apr 1;1861(4):148027. doi: 10.1016/j.bbabi.2019.05.003. Epub 2019 May 31. PMID: 31153887.

Latasa M, Cabello AM, Morán XAG, Massana R, Scharek R. Distribution of phytoplankton groups within the deep chlorophyll maximum. *Limnol Oceanogr.* 2017 Mar;62(2):665-685. doi: 10.1002/lno.10452.

Figure presentation and conceptual synthesis

A further limitation of the manuscript is the absence of a unifying summary figure that encapsulates the central conclusions of the study. While Figures 4 and 5 provide detailed structural comparisons, they primarily illustrate local differences in pigment arrangement and subunit organisation rather than conveying an overarching conceptual framework. As a result, the reader is left without a clear visual synthesis of how the reported structural observations collectively advance understanding of antenna evolution, excitation-energy-transfer reorganisation, or red-lineage diversification. For a manuscript positioned at the level of Nature Communications, a schematic or integrative figure that distils the main conceptual message would be essential.

We fully agree with this comment and appreciate the opportunity to communicate our conclusions, conceptual advances and hypotheses better. The detailed structural comparison previously shown in Fig 5 have now been moved to Supplementary Fig. 14. Importantly, this revised manuscript includes a unifying synthesis figure as Fig. 5. In it, we summarize the antenna subfamily differences across the organisms, highlight four identified drivers of antenna conservation and diversification, together with two additional hypothetical drivers. These drivers will provide the field with new concepts and hypotheses with which to analyze structures and derive insights, allow the field to determine the “so what?” of current and future structures beyond mere description.

As stated above, the recent findings in green endosymbiont *E. gracilis* suggests that the drivers we have identified are general to higher-order endosymbiosis and the architecture of photosystem antenna in complex chloroplasts, beyond the red lineage (Li et al, 2026).

Reference:

Li, K., Qin, BY., Zhang, YZ. et al. Structure and energy transfer of a far-red-absorbing euglenophyte PSI-LhcE-LhcbM supercomplex. *Nat Commun* (2026). <https://doi.org/10.1038/s41467-026-70067-1>

Fig. 5. Drivers of antenna conservation and diversification in the red lineage. **a** Schematic representation of PSI-FCP supercomplexes across the red lineage, based on currently available structures discussed in text. FCP subfamilies represented indicated by colours as in **(b)**. Symbols for drivers of antenna evolution overlaid on PSI-FCP as in **(c)**. For simplicity, belts are represented as full annuli rather than partial belts. **b** Schematic representation of FCP subfamilies present in each clade. **c** Summary of antenna evolution drivers and examples discussed in text. Converging arrows indicate driver for conservation; diverging arrows indicate driver for diversification. H1, H2, hypothetical drivers proposed in text.

Assignment and nomenclature of small PSI subunits

From the perspective of a non-specialist reader, the assignment and nomenclature of the PSI subunit referred to as PsaR raise a number of unresolved questions. In earlier work, including a 2008 study describing the purification and characterisation of *Acaryochloris* PSI, a small PSI-associated component was designated as Psa27. The introduction of a numerical designation at that time could reasonably be interpreted as reflecting a historical progression beyond the established letter-based naming scheme for PSI subunits. Against this background, it is not immediately clear why more recent structural studies have introduced or reintroduced letter-based designations, such as PsaR, for small PSI-associated components. For a reader without detailed prior knowledge of PSI subunit nomenclature, it is difficult to ascertain whether Psa27 and PsaR are intended to represent the same evolutionary or structural entity, lineage-specific variants of a related component, or fundamentally different classification concepts based on distinct criteria. This uncertainty is further compounded by the coexistence of multiple nomenclature schemes in the recent literature. While some studies, including an eLife report published in 2024, provide a historical and evolutionary framework for PSI subunits, other recent structural analyses adopt alternative assignments without clearly reconciling them with earlier designations. As a result, the continuity of PSI subunit nomenclature across the literature is not readily apparent to the reader. Clarification of the rationale underlying the present assignment—specifically, how it relates to previously defined components such as Psa27, and why a particular nomenclature scheme is preferred—would

therefore be highly beneficial. An explicit discussion of the criteria used for subunit identification, including sequence homology, structural correspondence, genomic context, and evolutionary conservation, would greatly improve transparency and facilitate comparison across studies.

Thank you for this important comment and the opportunity to clarify. We are not aware of the reasons why other authors instituted PsaR (Xu et al, October 2020) rather than the pre-existing Psa28 (Nagao et al, May 2020). We interpret it as Xu et al choosing PsaR as the first available “unclaimed” letter after PsaN/O/P/Q, likely without realizing Nagao’s designation as Psa28 given the similar timelines of paper submission/review/publication. We were not able to find out from the literature why Psa27 was chosen over the letter-based scheme when letters were still “available”. We fully agree that, to avoid ambiguities and confusion, it would be best to state the correspondence between the letter- and number-based systems going forward.

It was not our intention to overlook the number-based designations or the work that gave rise to it. We were under the wrong impression that our original manuscript contained the designation “PsaR (also known as Psa28)”, following on Kato et al. 2024 (eLife). This paper includes an excellent description and phylogenetic analysis of Psa28/R that informed many of our analyses. We apologize that this designation was inadvertently lost in version control before the original submission. We have now added this clarification and have made an analogous clarification for Psa29/PsaS (Xu et al, 2020) to improve transparency and facilitate comparison across studies. We prefer to use the PsaR designation such that we can maintain a letter-based system for PSI subunits in contrast to number-based system for FCPs, which also facilitates more streamlined figures.

References:

Xu, C., Pi, X., Huang, Y. et al. Structural basis for energy transfer in a huge diatom PSI-FCPI supercomplex. *Nat Commun* 11, 5081 (2020). <https://doi.org/10.1038/s41467-020-18867-x>

Nagao, R., Kato, K., Ifuku, K. et al. Structural basis for assembly and function of a diatom photosystem I-light-harvesting supercomplex. *Nat Commun* 11, 2481 (2020). <https://doi.org/10.1038/s41467-020-16324-3>

Kato, K., Nakajima, Y., Minoru, Xiang J., Kumazawa, M., Ogawa, H., Shen, J.-R., Ifuku, K., Nagao, R. (2024) Structural basis for molecular assembly of fucoxanthin chlorophyll *a/c*-binding proteins in a diatom photosystem I supercomplex. *eLife* 13:RP99858.

Scope and grounding of the concluding discussion

In the concluding part of the Discussion, the manuscript broadens its interpretation to encompass wider ecological and societal contexts, including the high photosynthetic productivity of kelp and the role of kelp forests as blue carbon ecosystems. While these themes are undoubtedly of considerable interest, their connection to the structural observations reported in this study is not sufficiently developed. In particular, it remains unclear how the structural features identified in the brown algal PSI-FCP complex—such as antenna architecture, FCP composition, or the proposed excitation energy transfer (EET) pathways—translate into demonstrable advantages in photosynthetic performance at either the organismal or ecosystem level. In the absence of direct physiological, biochemical, or ecological evidence, these broader interpretations appear largely aspirational rather than being firmly supported by the data presented. Consequently, the final section of the Discussion risks extending the implications of the structural analysis beyond what can be robustly justified, moving from well-supported comparative observations towards more speculative statements concerning productivity and carbon sequestration. A more cautious framing, or a clearer distinction between conclusions directly supported by the data and longer-term perspectives, would substantially enhance the coherence and credibility of the Discussion.

In the last two sentences of our original Discussion, we stated: “*This first cryoEM structure of a brown algal complex lays the groundwork to understand kelp’s high photosynthetic productivity ‘from the bottom up’, providing a biochemical basis for the certification of kelp forests as blue carbon ecosystems (26). Further structural comparisons across the red lineage, as well as biochemical, biophysical and genetic analyses are needed to derive the full complement of drivers shaping PSI-FCP architecture, composition and function.*”

We completely agree with the reviewer that the blue-carbon mention in the penultimate sentence of our Discussion is broad, forward-looking and aspirational. We also completely agree that it remains unclear how the structural features identified in the brown algal PSI-FCP complex translate into demonstrable advantages in the photosynthetic performance at either the organismal or ecosystem level. Whereas the “how” is unclear, it has been demonstrated that kelp forests are 2-3 times more productive per unit area than oceanic phytoplankton, showing comparable or higher rates than more terrestrial ecosystems and agricultural crops (Pessarrodona et al, 2022; see Table S2). Moreover, the significant contribution of kelp forests to blue carbon and global ecosystem services is increasingly recognized thanks to improved measurements and models (see Pessarrodona et al, 2022, Eger et al, 2023, Filbee-Dexter et al, 2020 and references therein).

As stated in the last sentence of the Discussion, a large number of structural, biochemical, biophysical, genetic (as well as physiological and ecological) studies are needed to develop the connection from the protein complexes to the whole organism, kelp population, holobiont, biome and human society. Our point is that primary productivity and global carbon sequestration outcomes that are normally measured at the macroscopic scale ultimately rest on the biochemical activity of the photosynthetic complexes. Our interest in the topic is linking the biochemical to the (increasingly measured) ecological from the bottom up, i.e., focusing on the biochemical function. Given that significant differences in primary productivity have been measured by others between kelp forests and other types of marine and land phototrophs (e.g., Pessarrodona et al, 2022), we respectfully disagree that the last two sentences of the Discussion make speculative statements that blur distinctions between conclusions directly supported by the data and longer-term perspectives.

We have edited the relevant sentence to: “*This first cryoEM structure of a brown algal complex lays the groundwork to understand kelp’s high photosynthetic productivity “from the bottom up”, to support the development of kelp forests as blue-carbon ecosystems.*”

Pessarrodona A, Assis J, Filbee-Dexter K, Burrows MT, Gattuso JP, Duarte CM, Krause-Jensen D, Moore PJ, Smale DA, Wernberg T. Global seaweed productivity. *Sci Adv.* 2022 Sep 16;8(37):eabn2465. doi: 10.1126/sciadv.abn2465. Epub 2022 Sep 14. PMID: 36103524; PMCID: PMC9473579

Eger, A.M., Marzinelli, E.M., Beas-Luna, R. et al. The value of ecosystem services in global marine kelp forests. *Nat Commun* 14, 1894 (2023). <https://doi.org/10.1038/s41467-023-37385-0>

Filbee-Dexter, K., Wernberg, T. Substantial blue carbon in overlooked Australian kelp forests. *Sci Rep* 10, 12341 (2020). <https://doi.org/10.1038/s41598-020-69258-7>

Overall evaluation

This work undoubtedly provides valuable comparative structural data and useful information for the specialist photosynthesis community. However, it does not establish a new general principle, a substantial conceptual advance, or a clear mechanistic or functional breakthrough. In its current form, the manuscript therefore appears better suited to a more specialised journal focused on structural or comparative photosynthesis research, rather than to a venue seeking broad conceptual impact.

Thank you for your detailed assessment. While our manuscript did not focus on establishing mechanistic or functional breakthroughs, we do present conceptual advances and new general principles (see first section for Reviewer #1). Our findings, synthesis and hypotheses are initial steps towards closing the inordinate gap in kelp bioenergetic biochemistry. They also present novel concepts and frameworks for the study and interpretation of photosynthetic complexes across red lineage evolution, providing broad conceptual impact. We are confident that, thanks to the revisions originating from the thoughtful comments of all reviewers, our advances and impact are much more clearly conveyed in this revised version.

Reviewer #2 (Remarks to the Author)

This is an interesting cryo-electron microscopy (cryo-EM) study of the photosystem I complex from the giant kelp *Macrocystis pyrifera*. The data identify differences in the chlorophyll network of the PSI-FCP supercomplex, as well as variations in the thickness of the transmembrane hydrophobic layer across the supercomplex, suggesting potential functional consequences for photochemistry. This work lays an important foundation for understanding the high photosynthetic productivity of kelp, reveals new factors contributing to the conservation and diversification of antenna systems, and provides insights into the evolutionary relationships among red-lineage organisms.

Overall, this is an excellent study performed by a top-tier group with strong expertise. However, it is not clearly described how the FCP proteins were identified and matched to specific sequences. Is the identification unambiguous, or are there remaining uncertainties?

Thank you for your kind words. Please see our comments on FCP assignment and confidence below.

Major comments:

1. The basis for identifying the FCP proteins is not clearly demonstrated, making it difficult to evaluate the assignments. In particular, the map quality of the second belt is insufficient, and the evidence supporting their assignment as Lhcf proteins is weak.

Thank you for your comment. While we took a lot of care in not over-assigning FCP sequence identities, this was not sufficiently explained in the original manuscript. We have re-organized the text to bring the FCP assignment discussion to the initial structural section, including mention of confidence and emphasizing that the resolution in the second belt is low, that family assignments for these FCPs are based on the best-fitting family per Q-score method and that assignment will need to be re-examined with higher resolution datasets. We provide new panels in Supplementary Fig. 10 (k-s) showing the validation of our Q-score methodology, as detailed in the “Non-sequence-assigned subunits” section below. We also note that we do not make functional claims for FCPs without sequence assignments.

“On the first belt, we assigned Lhcr, Lhcq, CgLhcr9-like (Cg9-l) and RedCAP subunits with high confidence. On the second belt, we assigned two subunits as Lhcr (FCP13) and Lhcf (FCP17) with high confidence. The resolution of six FCP focused-refined maps was insufficient to assign protein sequence (FCP2/15/16/19/A/B). Nevertheless, FCP subfamilies are known to show structural differences (6) (Supplementary Fig. 10a-e). For instance, Lhcf proteins have a shorter and straighter helix C than Lhcr and Lhcq proteins, providing useful features to differentiate subfamilies (Supplementary Fig. 10f-j). Thus, we reasoned it would be possible to identify the best subfamily match by calculating map-model-fit (Q-scores, Supplementary Fig. 10k-s) (34). Q-scores of assigned models and maps of different subfamilies fit with non-cognate partners showed that subfamily discrimination is possible even at ~5 Å resolution

(Supplementary Fig. 10k, l). Thus, we measured the Q-scores of assigned subfamily model representatives into each unassigned map (Supplementary Fig. 10m-r) and of our unassigned poly-ala models into representative maps of each subfamily (Supplementary Fig. 10s). These complementary measures identified Lhcf as the best-matching subfamily for FCP15/16/19/A given our current data, to be confirmed with higher resolution structures. The subfamilies for FCP2/B remained unassigned due to their poorly discriminating Q-scores.”

Sequence-assigned subunits: In the Methods we describe our process to assign FCP protein identity based on careful evaluation of amino acid-by-amino acid differences between the top candidates (top hits from diatom queries). An example of the evaluation is below, where all key amino acid differences between hits were marked at the sequence level and then the model/map fit was evaluated by two independent expert structural biologists (P. Maturana, M. Maldonado). Our map-based assignment expertise comes from previous experience identifying unannotated mitochondrial-complex proteins in *Tetrahymena thermophila*, with >35 new subunits in complex IV and >15 subunits in complex III₂ and complex I assigned from the map, as well as the evaluation of multiple isoforms in plant respiratory complex III₂ (Zhou et al, 2022). We have added a column to Table S2 where we indicate our confidence level for each FCP sequence assignment (intermediate/high). Overall, focused-map resolutions ~2.5-3.2 Å (FCP1/3/5/6/7/8/10/11/17) led to high confidence assignments and resolutions ~3.7-3.8 Å (FCP4/9/13) led to intermediate confidence assignments. We provide examples for representative assignment confidence below. Moreover, even for the intermediate-confidence assigned FCPs, the top candidates belonged to the same family as the assigned sequence. Therefore, even if the sequence assignment were corrected with future higher-resolution maps, our conclusions regarding antenna subfamily composition of the sequence-assigned subunits would *not* change.

FCP3 (high confidence):		FCP13 (intermediate confidence):			
9638407	MKSVI--ALTAVLASASAFVPTGPFVGSRVTSVARPASSSL----SMAAADM-----	46	9541138	MKTAFASVLMVGSTFAFVAPSAARALASS-NSLDTSRGSTTTRATPMSFAGGLRG	59
9760439	MKMEIGAMALCSLSAVTAFVAPSFTGSIARVCHTSATTKMAVNDMLGADVETDGVFDP	60	9447818	MKA-TVASFTALLASAFMAPPLSRTAAPSSSSVCMH----ANSKAIFFPMPQPEGLDG	55
	** * . *: : ** * . * : * : * : * : * : * : * : * : * : * : *		9608268	MVN-SLAVAAAFACASAFITPTPLARTAAPQSSGMITQ----AAKSKSLPFMPQPPALDG	56
			*	* : * : * : * : * : * : * : * : * : * : * : * : * : *	
9638407	-----ERTNEWRAEKHKGRVAMLAUVGFLIQENWHPLYNGSLSNPKKATFEVPPREG	99	9541138	ADGPEFNS----KNIDPLGIAAARPENLLEAREAEIKHGRIAMLAICGLTAPE-FVRVP	113
9760439	LGYGKDDASLFRRAVELKHGRVAVLAVTGYLFAEQWHPLYDGLSPGL-KALGELPFAA	119	9447818	SMFIVPSAAMRDEGVDA-TKS--VDTLVYMRAEELKHGRVAQLAVVGNILVDQGVRF	112
	* : * : * : * : * : * : * : * : * : * : * : * : * : *		9608268	TMAGD-----VGDPIGFS--NFIPLDLREAEELKHGRICQLAVGVFASTDLGLHLP	106
			:	: * : * : * : * : * : * : * : * : * : * : * : * : *	
9638407	ILQIVMIFGLEWVISQIKLTPGYTPGDYVGSSTDLF-DGGEEDSTHRNFKLKHNGRAA	158	9541138	GEIFENV-SVLDAHNVMEKGMVQLLEFISLAEVLLEPTVIDLMDKDKREPGDFALDPL	171
9760439	MVQILGAIIVIELTVGKQDYE-NKAPGELGNFGQSWNPYPDNPVAFSKLQKELKNGRLA	178	9447818	GAQYAAISQSDAHDPMVAAGNMTMLLGAFFLEMVGGAAIFGAASGSRAPGDFGMDPL	172
	: * : * : * : * : * : * : * : * : * : * : * : * : *		9608268	GAMHDV--SSVAAHDAVAASGAMPQILLWSAFEAISTVAAVQMLEGSRVPGDFGFDPL	164
	** * : * : * : * : * : * : * : * : * : * : * : * : *		*	* : * : * : * : * : * : * : * : * : * : * : * : *	
9638407	MMGISGLVTHNLITGGPAFEQISRGVYTG---GIQ--	191	9541138	NLCKTPE---KLERLKLAEKNGRLAMFAVSGALTQMAMTGHGFPPMA	216
9760439	MLAIMGENVQEALTGQT-AIEQLTSGHSLPFGDQGQFF	215	9447818	NLTSNPS---KKARFELSEIQHCLRAMMAISGIATQSVLNGGAFPPYTG	217
	* : * : * : * : * : * : * : * : * : * : * : * : *		9608268	GLYSKPTMAKKKASMEKKEITHCLAMLAFSGMVTQAVLTDGDFPPYTG	212
	* : * : * : * : * : * : * : * : * : * : * : * : *		*	* : * : * : * : * : * : * : * : * : * : * : * : *	
 Resolution: 3.1 Å Majority of non-conserved residues have clear match to 1st candidate (green) Several non-conserved residues from the 2nd candidate are a clear mismatch (red) Few non-conserved residues are ambiguous between 1st and 2nd candidate (yellow) Non-modelled residues in termini are shown in grey 		 Resolution: 3.8 Å Majority of non-conserved residues have clear match to 1st candidate (green) Several non-conserved residues from the 2nd candidate are a clear mismatch (red) Several non-conserved residues are ambiguous between 1st and 2nd candidate (yellow) Non-modelled residues in termini are shown in grey 			

Non-sequence assigned subunits: To validate our Q-score method for family assignment, we determined the Q-scores obtained from the following model/map pairs:

- Assigned models of different subfamilies fit into FCP17 map (the only sequence-assigned kelp Lhcf). We calculated Q-scores using the map at its cognate resolution (3.7 Å) as well as maps

filtered to lower resolutions equivalent to those seen in the non-assigned FCP maps (4.7 Å, 5 Å, 5.2 Å). This provides a baseline for the Q-scores that can be expected when non-Lhcf models are fit into an Lhcf map at resolutions 3.7-5.2 Å. As can be seen in Supplementary Fig. 10k (pasted below), there is a >60% drop in the Q-score between an Lhcf model and a non-Lhcf model when fit into an Lhcf map. Averages, standard deviations and t-tests cannot be performed given that there is only one kelp Lhcf assigned model. Lhcf models from other organisms (e.g., diatoms, dinoflagellates) are not appropriate either, as there is a significant difference in scores across organisms (not shown) that confounds the results.

- b) Assigned kelp FCP17 model (Lhcf) into maps of different assigned families (kelp Lhcr, Lhcq, CgLhcr9-like), at their cognate and lower resolutions. This provides a baseline for the fit between an Lhcf model fit into other-family maps. As can be seen in Supplementary Fig. 10l, there is a ~40% drop between the fit of an Lhcf model into an Lhcf map compared to the next-best family. Similarly to a), statistical tests could not be performed for families Lhcq and Cg9-l, as only one representative of each family exists in the current structure, and other organisms introduced counter-productive variability.

Together the above suggests that the cognate versus non-cognate FCP family can be reasonably differentiated even with >5 Å resolution maps. The above findings would be strengthened by measurements from additional representatives for each family, unavailable from the current structure. With this, we proceeded to test the fit of the unassigned kelp maps (FCP15/16/19/A/B/2), by measuring the Q-scores between the following model/map pairs:

- c) Assigned models of different families fit into each of the unassigned maps (FCP15/16/19/A/B/2), as summarized in Supplementary Fig. 10m and illustrated in Supplementary Fig. 10n-r.
- d) Unassigned, poly-ala models for FCP15/16/19/A into maps of different assigned families (kelp Lhcr, Lhcq, CgLhcr9-like, Supplementary Fig. 10s).

For all unassigned FCPs, Lhcf was the highest-scoring family, both when fitting all subfamily models into the unassigned maps (Supplementary Fig. 10m) and when fitting the poly-alanine models into maps of different subfamilies (Supplementary Fig. 10s), except for FCP2. Given the low discrimination of FCPB/2, we decided to leave family unassigned.

Moreover, in this revision we were able to improve the resolution of the focused map for FCPA from 4.73 Å to 3.97 Å, leading to an improved model. The Q-scores of the improved FCPA map confirmed subfamily Lhcf as the best fit, with an increased difference from the 2nd-best fit relative to the original FCPA map. This provides further confidence for our method and our preliminary assignment of FCP15/16/19 as Lhcf proteins. Ultimately, all these family assignments need to be confirmed by higher resolution maps for each FCP.

Supplementary Fig. 10 (partial legend). k-s Q-score validation (k, l) and test on unassigned maps (m-s). For clarity, representatives are labeled with the key letter for each family, followed by the FCP number, e.g., F (17) corresponds to subfamily Lhcf, using FCP17 as the representative. Subfamily colours used as in (a-j). Res, resolution. Given that only Lhcr subfamily contains more than one representative, the average and standard deviation are only shown for Lhcr. (k) Q-scores of subfamily models fit into FCP17 map at its cognate resolution (3 Å) and low-pass filtered to lower resolutions as indicated. (l) Q-scores of FCP17 model fit into subfamily representative maps at cognate (3.7, 3.1, 3.2-3.8, 3 Å for Lhcf, Cg9Lhcr-like, Lhcr, Lhcfq respectively) and filtered to indicated lower resolutions. (m) Q-scores of subfamily representative models and cognate poly-alanine model (FCP15/16/19/A/B/2) fit into maps of FCP15/16/19/A/B/2. (s) Q-scores of poly-alanine models of FCP15/16/19/A/B/2 fit into maps of assigned subfamily representatives, as well as to their cognate maps (e.g., model for FCPA fit into FCPA map).

References:

Zhou L*, Maldonado M*, Padavannil A, Guo F, Letts JA. Structures of Tetrahymena's respiratory chain reveal the diversity of eukaryotic core metabolism. *Science*. 2022 May 20;376(6595):831-839. doi: 10.1126/science.abn7747. Epub 2022 Mar 31. PMID: 35357889; PMCID: PMC9169680.

2. Although differences in membrane thickness are observed, the relationship between these differences and functional implications remains unclear.

Thank you for your comment. We agree that this was not very clear in the original manuscript. We have edited the manuscript to clarify two separate membrane-thickness-related effects below in the membrane rippling section. We also now call out these two effects in the new summary figure (Fig. 5).

In our view, there are two key effects related to hydrophobic mismatch. The first one is mismatch minimization, particularly between PSI and the first belt. We induce this effect from observations across the lineage, and we propose it explains the rotations that accompany the 1st-belt family switches. The

second effect is the generation of mismatch, which we speculate may be exploited to tune the absorbance properties of chromophores within particular antenna positions. In other words, the advantage of incorporating a particular LHC subfamily at a particular location may be due differences in transmembrane thickness between the subfamilies (and its consequences in the absorption of the chromophores) in addition to the chromophore stoichiometry differences. This effect would most likely be more prominent in the outer belts, which have fewer constraints from protein:protein interactions and are therefore easier to diversify. If this hydrophobic mismatch effect is being exploited, it also suggests CNE is not acting at that particular location (see CNE section for Reviewer #1), i.e., that mismatch could lead to positive rather than neutral diversification.

In the manuscript:

“We posit two main functional implications for the transmembrane differences in the antenna. First, the FCP1/3 rotations in the red lineage decrease the hydrophobic mismatch between neighbouring antenna proteins (Supplementary Fig. 13f-i). This is also the case in recently reported LHCI rotations in green secondary endosymbiont E. gracilis (41, 42) (Supplementary Fig. 13j, k). Thus, we propose hydrophobic-mismatch-minimisation as a general and parsimonious explanation for FCP/LHC rotations across photosynthetic antennae. Second, membrane thickness affects the membrane’s dielectric properties and impacts the absorption spectra of chromophores (44–46). From this, we infer that FCP subfamilies with different transmembrane thickness could have different absorption properties even with equivalent chromophore composition. To our knowledge, this has not been directly tested. We hypothesise that the thickness of the local membrane will affect FCP absorption and EET properties and thus could provide additional selective advantages for particular FCP subfamilies at specific antenna locations. Biophysical experiments with reconstituted PSI supercomplexes, among others, will shed light on these hypotheses.”

3. The FSC curve in Extended Data Fig. 2 reaches the Nyquist limit, suggesting that the resolution estimation may not be accurate. The data were collected using a Falcon 4i detector, and it should be possible to import the data with up-sampling. Doing so might further improve the resolution and potentially enhance the density quality of the peripheral (second-belt) FCPs.

Thank you for this helpful suggestion. Given that data were collected in .tif without super-resolution rather than in EER format, unfortunately effective up-sampling is not possible. Up-sampling our .tif data would entail interpolation/padding rather than super-resolution, and would thus not result in increased resolution beyond the physical Nyquist limit of the original pixels. Nevertheless, a re-examination of our particle picking training led to a larger initial clean particle set that, after our step-wise local refinements (Supplementary Fig. 3-6) resulted in an improved map and model for FCPA. This new FCPA map was incorporated into the updated composite map and deposited to EMDB (Supplementary tables 2-3).

Minor comments:

Extended Data Figures and Supplemental Figures are mixed, which may cause confusion and should be unified.

In this revised version, we have consolidated Extended Data and Supplementary figures into Supplementary figures only. We agree that this format is much clearer.

Reviewer #4 (Remarks to the Author)

I have enjoyed reviewing this manuscript by Weissman and Maturana from Maria Maldonado's lab. Overall the manuscript is well written and clearly explain its points and the figures are informative and visually appealing.

The study presents the first high-resolution structure of the PSI-FCP supercomplex from *M. pyrifera*, revealing a two-belt antenna organization distinct from diatoms and other red-lineage algae. The work provides detailed insights into FCP subfamily composition, chromophore arrangement, and potential functional implications for excitation energy transfer. The authors are also proposing constructive neutral evolution as a mechanism driving outer-belt diversification. The methodology is sound, analyses are appropriate and conclusions are well supported by the data. This study represents a significant advance in structural and evolutionary understanding of photosynthetic antennae, with relevance for photosynthesis research, evolutionary biology, and marine ecology. *M. pyrifera* forms dense kelp forests that are ecological powerhouses, fueling coastal food webs, shaping marine habitats and driving primary production on a massive scale. Clarifying the structure and function of giant kelp's photosynthetic machinery is therefore key to understanding the productivity and resilience of these vital ecosystems.

Minor editorial revisions (such as ensuring consistent figure numbering, correcting grammar and adding a few clarifications) would further improve readability, but these do not affect the overall quality or impact of the work.

Not clear sentences or mistakes:

Thank you for your very detailed reading and helpful suggestions.

- In the main text, the authors refer to "Supplementary Fig. X", whereas in the figure legend of the Supplementary file the figures are labeled as "Supplemental Fig. X". Please use consistent terminology throughout the manuscript.

Edited.

Page2:

Rephrase please:

11 Whereas photosystems are structurally and functionally conserved, antenna systems and the chromophores

12 they contain are highly diverse, tuned to optimal absorption of wavelengths in different

13 environments (6).

Edited to "Whereas photosystems are structurally and functionally conserved, their antenna systems and chromophores are highly diverse, tuned to absorb light in different environments."

Page3:

Rephrase please:

32 To understand drivers of conservation and diversification the antenna and shed light on

33 functional and evolutionary relationships between red lineage phyla,

Edited to "To understand drivers of antenna conservation and diversification and shed light on evolutionary relationships, we compared antenna architecture and FCP subfamily composition across the red lineage."

Page4:

Repetition & grammar:

3 The Lhc majority in the first belt of the *M. pyrifera* PSI antenna is seen is

4 seen across all the red lineage, and it is an absolute majority in red algae and cryptophytes ...

Edited to “The Lhc majority that is seen in *M. pyrifera*’s first belt is conserved across all the red lineage; this Lhc majority is absolute in red algae and cryptophytes.”

Repetition & grammar:

16 center (Extended data Fig. 7n,o). Indeed, fast transfer (<10 ps) between between these...

Fixed.

Rephrase please:

22 ...tree, the resolution of

23 FCP15/16/19/A was insufficient resolution to assign protein sequences.

Fixed.

Grammar:

40 ...switch and rotation can be parsimoniously explained by them resulting from of gene losses that

41 broke contingent...

Edited to “FCP1/3’s subfamily switch and rotation can be parsimoniously explained by gene losses that break contingent PSI:FCP binding interactions and allow the “empty” positions to be filled in new ways, as discussed below.”

Page5:

Repeat:

18 This decreases the difference in transmembrane helix height between between...

Fixed.

Page6:

Rephrase please:

33 Mismatch minimization also provides a

34 parsimonious explanation as to why FCP rotations are favored belt upon gene loss and subfamily

35 switches (Fig. 2).

Edited to “Mismatch minimization also provides a parsimonious explanation for why FCP rotations are favored upon gene loss and subfamily switches, as seen for FCP1/3”.

Page8:

Rephrase please:

14 Clear support for the Pietluch model comes from FCP9/13, where cryptophytes and haptophytes

15 are closely aligned, but different from ochrophytes and dinoflagellates, themselves closely

16 aligned (10) (Fig. 1c).

Edited to “The Pietluch model is strongly supported by the FCP9/13 structures, which show two distinct clusters: cryptophytes/haptophytes and ochrophytes/dinoflagellates.”

Questions and suggestions:

General observations:

- Several “Extended Data Fig. X” figures are either not cited (Extended Data Fig. 1) in the main text or are referenced in a non-sequential order (e.g., Extended Data Fig. 2-5 are the first to be mentioned). Please ensure that all Extended Data figures are cited in the appropriate sections of the text and follow a logical numbering order.

Thank you. “Extended Data Figs. 2-5” should have read “1-5”. Some Extended Data figures were cited in the supplementary text only, leading to confusion. In this revised version, we have consolidated Extended Data and Supplementary figures into Supplementary figures only and ensured all figures are cited in fully sequential manner.

Page3:

13 “After optimizing protocols to decrease the viscosity of the kelp sample, we isolated a
14 chloroplast-enriched fraction from fresh *M. pyrifera* blades, from which we extracted and
15 purified PSI-FCP for structural determination using cryogenic electron microscopy (cryoEM)
16 (Extended Data Figs. 2-5, Supplementary table 1-3).” —> Please mention the detergent used in this purification, as it is one of the differences from previously published structures.

Thanks for the suggestion. Edited sentence to include digitonin.

37 “Note that the FCP number refers to the position in the antenna, not to the gene
38 name. For clarity, all antenna proteins are referred to as “FCP” using our numbering, even if the
39 LHC subunit binds other chromophores in certain organisms. —> It would be helpful to include a table that links each FCP discussed in this manuscript to the corresponding FCPs in other organisms, facilitating comparison with existing nomenclature. This table would support what shown in Supplementary figure 4

Thanks for this helpful suggestion. We have added a table to Supplementary Fig. 12.

Page4:

1 “Similarities in sequence, subunit composition and architecture of the supercomplex are strongest
2 in PSI, weakening with increasing distance from the core.” —>I would recommend mentioning from the beginning whether the PSI core is conserved across brown algae.

Thank you for pointing this out. This point was lost at some point between manuscript iterations. We have added the following sentence to the first paragraph of section “*M. pyrifera* PSI-FCP structure and FCP phylogeny”:

“PSI subunits were highly conserved with respect to the red lineage. The structure also confirmed the lack of PsaK in brown algal PSI (present in red algae, cryptophytes and haptophytes), as well as the lack of PsaO (present in red algae and cryptophytes).”

14 “This Lher-specific chlorophyll is ~6 Å away from chl 407 of the adjacent FCP, poising
15 these chlorophyll molecules for efficient excitation energy transfer (EET) into PSI’s reaction
16 center (Extended data Fig. 7n,o)”. —> The authors state that a ~6 Å distance is conducive to efficient
excitation energy transfer. It would be helpful to specify the distance range typically required for efficient
EET, to better support this interpretation.

Distances of ~10 Å show strong excitonic interactions and lead to fast EET. We have added the below
PSI-EET reviews as references for our statement.

Croce R & van Amerongen H. Light harvesting in oxygenic photosynthesis: Structural biology meets
spectroscopy. *Science* 369, eaay2058(2020). DOI:10.1126/science.aay2058

Croce R, van Amerongen H. Light-harvesting in photosystem I. *Photosynth Res.* 2013 Oct;116(2-3):153-
66. doi: 10.1007/s11120-013-9838-x. Epub 2013 May 4. PMID: 23645376; PMCID: PMC3825136.

28 “The subfamilies of FCPB, as well as FCP2 in the first belt, remained unassigned.” —> It would be
helpful to clarify in which organisms FCPB proteins have previously been assigned to specific
subfamilies, if any. This would help place the present limitation in a broader comparative context.

Thank you for this helpful point. The kelp structure is the first one in which a protein is seen in the FCPB
(or FCPA) position, but this information was in the supplementary text only. We have now added this
detail to the main text:

*“We also note that FCPA/B are new antenna positions not previously seen in other organisms (13, 14,
73, 74, 15–24).”*

As it has not previously been seen in other organisms, no subfamilies have yet been assigned to it in any
context.

Materials and Methods

- How was assessed the purity and quality of the sample?

Sucrose-gradient fractions were selected based on their maximum absorbance peak at 676–678 nm and
processed for cryoEM grid preparation. In line with our previously validated cryoEM workflow for
mitochondrial electron-transport chain complexes (see references below), we do not further purify the
sample biochemically. Rather, we use partially purified (i.e., biochemically heterogeneous) samples as the
cryoEM grid material and do “*in silico* purification” by selecting and iteratively cleaning the relevant
particles using cryoEM 2D and 3D classification algorithms. This minimizes the time and extent of
biochemical manipulation, protecting the membrane complexes from degradation and sample loss. In the
case of the PSI sample, the grid also included low numbers of particles of other photosynthetic membrane
complexes, which were removed from downstream cryoEM processing in the early stages.

References:

Maldonado M, Padavannil A, Zhou L, Guo F, Letts JA. Atomic structure of a mitochondrial complex I
intermediate from vascular plants. *Elife.* 2020 Aug 25;9:e56664. doi: 10.7554/eLife.56664. PMID:
32840211; PMCID: PMC7447434.

Maldonado M, Guo F, Letts JA. Atomic structures of respiratory complex III₂, complex IV, and supercomplex III₂-IV from vascular plants. *Elife*. 2021 Jan 19;10:e62047. doi: 10.7554/eLife.62047. PMID: 33463523; PMCID: PMC7815315.

Maldonado M, Fan Z, Abe KM, Letts JA. Plant-specific features of respiratory supercomplex I + III₂ from *Vigna radiata*. *Nat Plants*. 2023 Jan;9(1):157-168. doi: 10.1038/s41477-022-01306-8. Epub 2022 Dec 29. Erratum in: *Nat Plants*. 2023 Mar;9(3):501. doi: 10.1038/s41477-023-01373-5. PMID: 36581760; PMCID: PMC9873571.

Zhou L, Maldonado M, Padavannil A, Guo F, Letts JA. Structures of *Tetrahymena*'s respiratory chain reveal the diversity of eukaryotic core metabolism. *Science*. 2022 May 20;376(6595):831-839. doi: 10.1126/science.abn7747. Epub 2022 Mar 31. PMID: 35357889; PMCID: PMC9169680.

- Which magnification was used for data collection?

The nominal magnification for our 1.226 Å/pixel was 105,000x.

Looking at the maps:

- Extended Data Fig.2

Looking at Extended Data Fig. 2, it would be helpful to show the distribution of particles that contributed to the final full map. Do the particles exhibit a preferred orientation? Including a particle distribution plot for the final map, as well as a local resolution map, would greatly help in assessing the quality of the map.

In Supplementary Fig. 2, we now include the particle distribution plot and the local resolution map of the full PSI-FCP map derived from ~165,000 particles, which is the last map before the masked focused refinements with the PSI mask and each FCP mask. Please note that the focused refinements significantly improved the resolutions of each masked region, as described in the rest of the figure for PSI and in Supplementary Fig. 3-6 for the antenna.

There is some preferred orientation for the membrane view relative to the “top” and “bottom” (stromal and luminal) views. This is equivalent for that seen for most, if not all, previous PSI-antenna supercomplexes in the red and green lineages. To minimize the preferred orientation and increase the representation of top and bottom view particles, we performed several iterative rounds of particle training, picking and classification.

As indicated by the local resolution heatmap, there is a difference in resolution between PSI (~2.7 Å-3.5 Å), first belt (~3 Å-5 Å) and second belt (~4.5 Å-6.5 Å) in the unmasked map with the full ~165,000 particle set. Therefore, as described in Supplementary Fig. 3-6, the set of ~133K particles was further classified and masked to improve the resolution of PSI and each individual FCP. This significantly improved resolution of the all FCPs (3 Å-3.8 Å for the first belt; 3.7 Å-5.21 Å for the second belt, Supplementary Table 2), as shown in the individual FSC curves (Supplementary Fig. 3-6). We have deposited a composite map as well as each individual FCP map (with their half maps and FSC curves) as per Supplementary Table 2.

- Extended Data Fig. 3-4-5

Some FCPs in the first belt and all FCPs in the second belt are observed only in subsets of particles. Could the authors comment on whether this reflects the purification conditions (light/dark), the detergent used, or possibly the functional dynamics of the supercomplex, such as transient assembly under particular physiological states?

This is a very interesting question that we are planning to investigate in more detail. Given that the extraction at lower detergent:ratio resulted in PSI-FCP supercomplexes with larger antennae (Supplementary Fig. 8), we presume that a large component of the current variability is due to biochemical purification conditions. Given that light-dependent changes in antenna architecture have been observed in diatoms, we hypothesize that analogous changes due to environmental exposures, as well as due to life cycle stage, exist in brown algae.

References:

Feng Y, Li Z, Yang Y, Shen L, Li X, Liu X, Zhang X, Zhang J, Ren F, Wang Y, Liu C, Han G, Wang X, Kuang T, Shen JR, Wang W. Structures of PSI-FCPI from *Thalassiosira pseudonana* grown under high light provide evidence for convergent evolution and light-adaptive strategies in diatom FCPIs. *J Integr Plant Biol.* 2025 Apr;67(4):949-966. doi: 10.1111/jipb.13816. Epub 2024 Dec 13. PMID: 39670505.